# CoPur: Certifiably Robust Collaborative Inference via Feature Purification

**Jing Liu**
Department of Computer Science
University of Illinois at Urbana Champaign
jil292@illinois.edu

**Chulin Xie**
Department of Computer Science
University of Illinois at Urbana Champaign
chulinx2@illinois.edu

**Oluwasanmi O Koyejo**
University of Illinois at Urbana Champaign
& Stanford University & Google Research
sanmi@stanford.edu

**Bo Li**
Department of Computer Science
University of Illinois at Urbana Champaign
lbo@illinois.edu

## Abstract

Collaborative inference leverages diverse features provided by different agents (e.g., sensors) for more accurate inference. A common setup is where each agent sends its embedded features instead of the raw data to the Fusion Center (FC) for joint prediction. In this setting, we consider inference phase attacks when a small fraction of agents is compromised. The compromised agent either does not send embedded features to the FC or sends *arbitrary* embedded features. To address this, we propose a certifiably robust COllaborative inference framework via feature PURification (CoPur), by leveraging the block-sparse nature of adversarial perturbations on the feature vector, as well as redundancy across the embedded features (by assuming the overall features lie on an underlying lower dimensional manifold). We theoretically show that the proposed feature purification method can robustly recover the true feature vector, despite adversarial corruptions and/or incomplete observations. We also propose and test an untargeted distributed feature-flipping attack, which is agnostic to the model, training data, label, as well as features held by other agents, and is shown to be effective in attacking state-of-the-art defenses. Experiments on ExtraSensory and NUS-WIDE datasets show that CoPur significantly outperforms existing defenses in terms of robustness against targeted and untargeted adversarial attacks.

## 1 Introduction

Collaborative inference is an increasingly popular distributed prediction strategy that leverages diverse features provided by different agents (e.g., sensors) for more accurate inference. For example, in context-aware music recommendation, the app may wish to recommend more appropriate music based on the user's behavioral context (e.g., sleeping, running) (Vaizman et al., 2017). Here, it can leverage diverse sensors from the user's smart phone and smart watch, e.g., phone accelerometer, gyroscope, watch accelerometer, location, audio, phone states (e.g., wifi, ringing mode), and environment sensing (e.g., light). Thus, by combining information across sensors, much more accurate predictions can be made. In this work, we study the scenario where Internet-of-Things (IoT) devices send embedded features (instead of the raw data) to a Fusion Center (FC) for collaborative inference. The IoT devices do not communicate with each other and may hold quite different raw data (e.g., image, acoustic, Lidar data, etc). Nevertheless, as a result of distributed inference, the system is vulnerable to attack, where a subset of sensors are corrupted by an adversary at inference time, e.g., due to the lack of effective security mechanisms, connection errors, sensor failures, among other issues – this is also a

36th Conference on Neural Information Processing Systems (NeurIPS 2022).

common concern in defense applications (Abdelzaher et al., 2018; Xie et al., 2020). Nevertheless, the goal of the system is robust inference based on the possibly incomplete and corrupted embedded features received from the IoT devices.

There have been a number of works on robust parameter estimation in IoT settings, where the robustness apparently comes from redundancy among different sensors' linear measurements of the same parameter. For a comprehensive overview, we refer the interested reader to (Zhang et al., 2018). However, in our considered inference setting, each IoT device may hold quite different types of features, and the redundancy is less apparent. We also observe that IoT applications are often communication-limited (Ko et al., 2018), thus, using lower-dimensional embeddings to communicate from each sensor is a common and effective approach to reduce the required communication overhead.

The problem we study here also applies to the inference phase of some Vertical Federated Learning (VFL) frameworks (Chen et al., 2020; Liu et al., 2020), where a server similarly combines different embedded features received from every agent to make a prediction. One difference between our setting and VFL is: the agents in VFL are other untrusted parties, if an agent in VFL is malicious, it will likely attack during both training and inference; while in our setting, the agents (e.g., IoT sensors) are all owned by FC, and they can be well protected during training, and inference is more vulnerable. So far, there are no certifiable defenses for VFL against adversarial attacks during the inference phase. The well-known robust aggregation method (Yin et al., 2018; Guerraoui et al., 2018; Blanchard et al., 2017; Fu et al., 2019; Pillutla et al., 2019; Fung et al., 2020; Chen et al., 2017; Xie et al., 2021) used for Horizontal Federated Learning (HFL) is not applicable.

Other related works will be discussed in Section 4. We use the term 'device' and 'agent' interchangeably in this paper. The main difficulty of defending against adversarial attacks in our collaborative inference setting (as well as in VFL) is that there is no apparent redundancy across the agents. Though, intuitively, there is some mutual information among the features held by each agent. Even worse, the raw data held by the agents are not accessible by the FC/server to inspect, the attacker can perturb the compromised agents' embedded features with a very large magnitude before sending them to the FC/server.

In this paper, we propose to improve robustness based on the underlying redundancy among the features held by IoT devices. Specifically, we assume the combined features lie on an underlying lower dimensional manifold. We propose a robust COllaborative inference framework via feature PURification (CoPur), which is able to defend against many types of attacks during inference (see the attack taxonomy in Section 2.2). During the inference, the FC receives the corrupted and/or incomplete embedded features from agents (Figure 1a). Then CoPur purifies the embedded features based on the learned feature subspace (Figure 1b). Finally, it feeds the purified features to FC's trained global model for prediction. The detailed robust inference procedures will be described in Sections 2.3. We aim to answer the following questions: *Can we make robust predictions based on corrupted and/or incomplete features? If yes, how many compromised devices can we tolerate?*

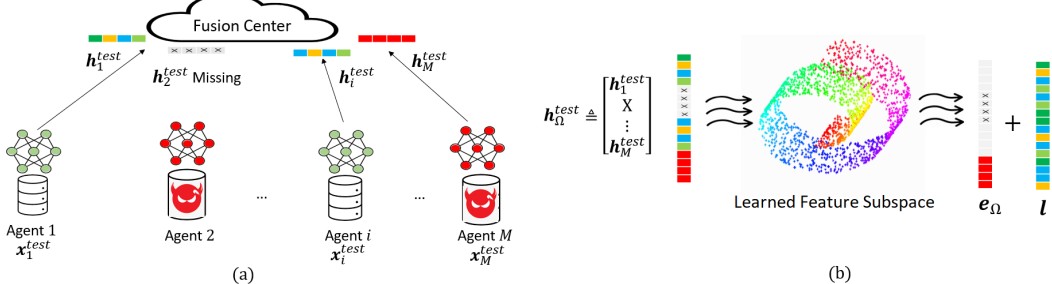

Figure 1: (a) Fusion Center receives corrupted and incomplete embedded features; (b) Feature Purification.

**Technical Contributions**: Our main contributions include the following:

- We propose a certifiably robust collaborative inference procedure to defend against a variety of targeted or untargeted attacks (e.g., distributed adversarial attacks, missing-feature attack, and their combinations) during collaborative inference.
- We propose a novel non-linear robust decomposition method that decomposes the potentially corrupted and incomplete feature vector into two parts: one lies (exactly or approximately) on the underlying feature manifold; the other with a block-sparse structure. Theoretically, we prove

that the proposed method can recover the true feature vector *exactly* (or *approximately*) despite corruptions and incomplete observations. This may be of independent interest.

- We propose and test a distributed feature-flipping attack, which is agnostic to the model, training data, label, as well as features held by other agents. Empirical studies show that this new attack effectively attacks existing state-of-the-art defenses.
- We conduct extensive experiments on ExtraSensory and NUS-WIDE datasets, and show that CoPur is significantly more robust than baselines against different types of attacks.

## 2 Robust collaborative inference via feature purification (CoPur)

We describe the system setting, the threat models, and the proposed robust inference procedures.

### 2.1 Collaborative inference and system setting

We first describe the basic framework of collaborative inference, which is also similar to the inference phase of some VFL frameworks (Chen et al., 2020; Liu et al., 2020). There are $M$ agents, where each agent $i$ holds partial raw data $\boldsymbol{x}_i^{test}$ of the overall testing sample $\boldsymbol{x}^{test} = [\boldsymbol{x}_1^{test}; ...; \boldsymbol{x}_M^{test}]$. Also, each agent $i$ has its own local feature extractor $f_i$ parameterized by $\theta_i$ that maps its raw data vector $\boldsymbol{x}_i^{test}$ to the embedded feature vector $\boldsymbol{h}_i^{test} \triangleq f_i(\boldsymbol{x}_i^{test}; \theta_i)$. The dimension of $\boldsymbol{h}_i^{test}$ is usually smaller than $\boldsymbol{x}_i^{test}$. The FC only receives the embedded features $\boldsymbol{h}_i^{test}, i = 1, ..., M$ from the agents (not the raw data $\boldsymbol{x}_i^{test}$), and concatenates them into a long column vector $\boldsymbol{h}^{test} \triangleq [\boldsymbol{h}_1^{test}; \boldsymbol{h}_2^{test}; ...; \boldsymbol{h}_M^{test}]$. Then the FC uses its pre-trained global model $f_{\theta_0}$ to make the prediction.

While we do not focus on training, we will require an AutoEncoder, trained on uncorrupted data (e.g., during training) that can approximately capture the manifold of the uncorrupted embedded features, *i.e.*, $\sum_{i=1}^{M} \|[\mathcal{D}_\phi(\mathcal{E}_\psi(\boldsymbol{l}^*)) - \boldsymbol{l}^*]_i\|_2 \leq \delta$ for the true embedded features $\boldsymbol{l}^*$ of the testing instance, where $\mathcal{D}_\phi$ is the decoder parameterized by $\phi$, and $\mathcal{E}_\psi$ is the encoder parameterized by $\psi$. Further, we assume that the global model for inference $f_{\theta_0}$ is based on the output of the AutoEncoder and the training labels, *i.e.*, $\{\mathcal{D}_\phi(\mathcal{E}_\psi(\boldsymbol{h}^{(j)})), y^{(j)}\}_{j=1}^{n}$. Detailed examples of how to train such a model and the AutoEncoder can be found in Section D of the supplemental material.

### 2.2 Threat model

There are $M$ agents which hold different parts of the feature of the same set of $n$ training instances. We assume the training process is well-protected (attack-free). However, during the inference, the attacker can attack $\alpha M$ agents. The compromised agent either is not able to send features to the FC, or sends arbitrary embedded features to FC.

We use $\boldsymbol{h}_{benign}$ to denote the overall embedded features provided by the $(1 - \alpha)M$ benign agents (these agents are indexed by $\Omega_{benign}$ and $|\Omega_{benign}| = (1 - \alpha)M$ ), and we use $\boldsymbol{h}_{adv}$ to denote the overall embedded features sent by compromised agents (indexed by $\Omega_{adv}$). The set $\Omega = \Omega_{benign} \cup \Omega_{adv}$ is the index set of agents that provide features, and its complement, *i.e.*, $\Omega^c$, denotes the compromised agents which do not provide features. We categorize the inference phase attacks into the following three types:

**Threat model A (distributed adversarial attack):** All the compromised agents jointly send adversarial embedded features $\boldsymbol{h}_{adv}$, with the goal to mislead the prediction to the target label $y^A$:

$$\min_{\boldsymbol{h}_{adv}} \ell(f_{\theta_0}(\{\boldsymbol{h}_{adv}, \boldsymbol{h}_{benign}\}), y^A), \tag{1}$$

**Threat model B (missing-feature attack):** All the compromised agents do not send any feature to the FC. A such missing-feature attack may be due to damaged sensors or disconnection, and can still affect FC's prediction. The presence of a missing-feature attack can be viewed as the FC observing only partial blocks (indexed by $\Omega$) of the overall embedded features $\boldsymbol{h}$, *i.e.*, $\boldsymbol{h}_\Omega$. $\Omega^c$ denotes the index set of agents that perform missing-feature attacks. The attacker can compromise up to $\alpha M$ agents to perform such an attack:

$$\min_{\Omega^c} \ell(f_{\theta_0}(\boldsymbol{h}_\Omega), y^A), s.t. \ |\Omega^c| \leq \alpha M \tag{2}$$

**Threat model C (combined attack):** The attacker can also perform a combined adversarial and missing feature attack by providing $\boldsymbol{h}_{adv}$, indexed by $\Omega_{adv}$, and missing features $\Omega^c$:

$$\min_{\Omega^c, \boldsymbol{h}_{adv}} \ell(f_{\theta_0}(\{\boldsymbol{h}_{adv}, \boldsymbol{h}_{benign}\}), y^A), s.t. \ |\Omega^c| + |\Omega_{adv}| \leq \alpha M \tag{3}$$

Beyond these targeted attacks, the attacker can also perform untargeted attacks by maximizing the distance between the prediction and the ground truth label.

## 2.3 Inference procedures of CoPur

We propose an all-in-one solution to defend against various inference phase attacks defined in Section 2.2. Let $\Omega$ be the index set of the observed blocks. The observed $h_\Omega$ can be decomposed as $h_\Omega = l_\Omega + e_\Omega$ where $l$ is the underlying uncorrupted features, and $e_\Omega$ models the corruptions on the observed set $\Omega$. Finding such a decomposition seems hard, even if the learned AutoEncoder can capture the underlying feature subspace, *i.e.*, $l = \mathcal{D}_\phi(\mathcal{E}_\psi(l))$. Fortunately, as we mentioned in Section 4, $e_\Omega$ has a block-sparse structure. We propose to solve the following:

$$\min_l \sum_{i \in \Omega} \mathbb{1}\{[h - l]_i \neq 0\} \quad s.t. \quad \mathcal{D}_\phi(\mathcal{E}_\psi(l)) = l, \tag{4}$$

where the indicator function $\mathbb{1}\{\cdot\}$ counts whether the block $[h - l]_i$ is nonzero. This block-sparsity minimization shares the same spirit as the $\ell_0$ minimization in robust linear regression (Candes & Tao, 2005; Mitra et al., 2013). However, it is NP-hard to solve, so we relax it to the $\ell_{2,1}$-norm:

$$\min_l \sum_{i \in \Omega} \|[h - l]_i\|_2 \quad s.t. \quad \mathcal{D}_\phi(\mathcal{E}_\psi(l)) = l \tag{5}$$

Our theoretical analysis in Theorems 1 & 2 shows that under certain conditions, solving the above relaxed objective is able to recover the underlying true features exactly.

But what if the learned AutoEncoder can only approximately reconstruct the underlying features, say $\sum_{i=1}^{M} \|[\mathcal{D}_\phi(\mathcal{E}_\psi(l)) - l]_i\|_2 \leq \delta$ ? In this case, we further relax the constraints in Eq. 5 and solve:

$$\min_l \sum_{i \in \Omega} \|[h - l]_i\|_2 \quad s.t. \quad \sum_{i=1}^{M} \|[\mathcal{D}_\phi(\mathcal{E}_\psi(l)) - l]_i\|_2 \leq \delta \tag{6}$$

Our theoretical analysis in Theorems 3 & 4 show that under certain conditions, solving Eq. 6 is able to approximately recover the underlying true features, despite the corruptions and/or incomplete observations. In our implementation, we first try to find a good initial point by simply searching the embedded features on the manifold, i.e., $\min_{l'} \sum_{i \in \Omega} \|[h - \mathcal{D}_\phi(\mathcal{E}_\psi(l'))]_i\|_2$ via gradient descend (It is useful to notice that since Eq. 6 allows $l$ approximately lie on the manifold, its objective value will be slightly smaller than $\min_{l'} \sum_{i \in \Omega} \|[h - \mathcal{D}_\phi(\mathcal{E}_\psi(l'))]_i\|_2$). Then we apply Lagrange Multiplier method to solve Eq. 6, i.e., $\hat{l} = \arg\min_l \sum_{i \in \Omega} \|[h - l]_i\|_2 + \tau \sum_{i=1}^{M} \|[\mathcal{D}_\phi(\mathcal{E}_\psi(l)) - l]_i\|_2$. More implementation details can be found in the supplemental.

After obtaining $\hat{l}$, we feed $\mathcal{D}_\phi(\mathcal{E}_\psi(\hat{l}))$ to FC's trained global model for the prediction.

## 3 Theoretical analysis of CoPur

We first analyze the 'noiseless case' where the learned AutoEncoder satisfies $\mathcal{D}_\phi(\mathcal{E}_\psi(l^*)) = l^*$ for the underlying uncorrupted feature vector $l^*$. Next, we analyze the more challenging 'noisy case' where the AutoEncoder can only approximately reconstruct $l^*$, *i.e.*, $\sum_{i=1}^{M} \|[\mathcal{D}_\phi(\mathcal{E}_\psi(l^*)) - l^*]_i\|_2 \leq \delta$. Proofs and additional discussions can be found in Section A of the supplemental material.

### 3.1 Noiseless Case

To simplify the exposition, we first consider threat model B, *i.e.*, the missing-feature attack.

**Theorem 1. (Exact feature recovery under threat model B)** Assume the trained AutoEncoder satisfies $\mathcal{D}_\phi(\mathcal{E}_\psi(l^*)) = l^*$ for the underlying uncorrupted feature vector $l^*$. For $\forall r', r''$ in the range of the decoder $\mathcal{D}_\phi(\cdot)$ where $r' \neq r''$, if $(r' - r'')_\Omega \neq 0$, then given $h_\Omega = l^*_\Omega$, $l^*$ is the unique solution of Eq. 5 and Eq. 4.

The proof can be found in Section A.3 of the supplemental material, where we show that any feasible point that is different from $l^*$ would have a larger objective value.

The condition $(r' - r'')_\Omega \neq 0$ required by Theorem 1 is intuitive and necessary. Suppose there are $r' \neq r''$ but $r'_\Omega = r''_\Omega$, then observing $r'_\Omega$ is not able to tell whether it's from $r'$ or $r''$. In general, when the observed set $\Omega$ gets larger, the required condition has a better chance of being satisfied. If we have complete observations, *i.e.*, $\Omega = \{1, 2, ..., M\}$, the required condition $(r' - r'')_\Omega \neq 0$ *automatically holds.*

Now we consider the more challenging case where we not only have missing features but also have corruptions on the observed features.

**Theorem 2. (Exact feature recovery under threat model C)** Assume the trained AutoEncoder satisfies $\mathcal{D}_\phi(\mathcal{E}_\psi(\boldsymbol{l}^*)) = \boldsymbol{l}^*$ for the underlying uncorrupted feature vector $\boldsymbol{l}^*$. For $\forall \boldsymbol{r}', \boldsymbol{r}''$ in the range of the decoder $\mathcal{D}_\phi(\cdot)$ where $\boldsymbol{r}' \neq \boldsymbol{r}''$, define $\boldsymbol{v} = \boldsymbol{r}' - \boldsymbol{r}''$, if for any partition $\{S_\Omega, \bar{S}_\Omega\}$ of $\Omega$ with $|\bar{S}_\Omega| = q < |\Omega|/2$, it holds that $\sum_{i \in S_\Omega} \|\boldsymbol{v}_i\|_2 > \sum_{i \in \bar{S}_\Omega} \|\boldsymbol{v}_i\|_2$, then for any $\boldsymbol{h}_\Omega = \boldsymbol{l}_\Omega^* + \boldsymbol{e}_\Omega^*$ with $\sum_{i \in \Omega} \mathbb{1}\{\boldsymbol{e}_i^* \neq \boldsymbol{0}\} \leq q$, $\boldsymbol{l}^*$ is the unique solution of Eq. 5 and Eq. 4.

The proof can be found in Section A.4 of the supplemental material. We first show that any feasible point of Eq. 5 that is different from $\boldsymbol{l}^*$ would have a larger objective value. Then we use contradiction to show that there does not exist any global optimal solution of Eq. 4 that is different from $\boldsymbol{l}^*$.

**Remark 1.** Theorem 2 not only requires $\boldsymbol{v}_\Omega \neq \boldsymbol{0}$ like Theorem 1, but also requires $\sum_{i \in S_\Omega} \|\boldsymbol{v}_i\|_2 > \sum_{i \in \bar{S}_\Omega} \|\boldsymbol{v}_i\|_2$. This is due to the presence of corruptions in the observed blocks (indexed by $\bar{S}_\Omega$). When there is no corruption, *i.e.*, $q = 0$, the corresponding set $\bar{S}_\Omega$ is empty and the required condition reduces to $\sum_{i \in \Omega} \|\boldsymbol{v}_i\|_2 > 0$, *i.e.*, $\boldsymbol{v}_\Omega \neq \boldsymbol{0}$. Second, Theorem 2 is a universal certification, where any subset $\bar{S}_\Omega$ of the observations can be malicious, as long as $|\bar{S}_\Omega| = q$. In practice, once the attacker chooses $q$ agents to perform adversarial attacks, the set $\bar{S}_\Omega$ gets fixed and corresponds to $\boldsymbol{e}_i^* \neq \boldsymbol{0}$. Then, we only require $\sum_{i \in S_\Omega} \|\boldsymbol{v}_i\|_2 > \sum_{i \in \bar{S}_\Omega} \|\boldsymbol{v}_i\|_2$ to hold for this particular fixed set $\bar{S}_\Omega$, to guarantee $\boldsymbol{l}^*$ is the unique solution of Eq. 5. This can be easily seen from the proof of Theorem 2. Third, the condition $\sum_{i \in S_\Omega} \|\boldsymbol{v}_i\|_2 > \sum_{i \in \bar{S}_\Omega} \|\boldsymbol{v}_i\|_2$ prefers $\boldsymbol{v}_\Omega$ to be not 'spiky'. Suppose $\boldsymbol{r}'$ and $\boldsymbol{r}''$ are the same on all the observed blocks (indexed by $\Omega$) except one, which means only one block of $\boldsymbol{v}_\Omega$ is non-zero (*i.e.*, $\boldsymbol{v}_\Omega$ is spiky). If that non-zero block is in the set $\bar{S}_\Omega$, the required condition will not be met. Intuitively, if there are two instances differing in only one block of the observed features, and the attacker corrupts that block, the FC can not recover the original feature. In general, we prefer $\boldsymbol{v}_\Omega$ to be not 'spiky', thus the condition can hold when the number of corrupted blocks $|\bar{S}_\Omega|$ is sufficiently small. Such condition can be considered as the non-linear and missing-block extensions of the Range Space Property in robust linear regression literature (Flores, 2015; Liu et al., 2018), which we will discuss next. Finally, the fraction ($q/M$) of corrupted blocks that CoPur can tolerate not only depends on $\Omega$, but also depends on the underlying feature subspace. It can tolerate up to $50\% \times |\Omega|$ corrupted blocks if there is enough redundancy among the agents, e.g., if every agent holds exactly the same embedded features, then every block of $\boldsymbol{v} = \boldsymbol{r}' - \boldsymbol{r}''$ is the same, and $q$ approaches $50\% \times |\Omega|$.

**Remark 2.** To draw the connections to the Range Space Property in robust linear regression, let's consider a typical case where there is no non-linear activation function in the last layer of the AutoEncoder (other layers can have non-linear activation functions). Let $\boldsymbol{W}_{end}$ be the weight matrix of the last layer of the AutoEncoder. Note that the difference between any two vectors from the range of the decoder $\mathcal{D}_\phi(\cdot)$ is contained in the $Range(\boldsymbol{W}_{end})$, *i.e.*, $\boldsymbol{v} \in Range(\boldsymbol{W}_{end}) \backslash \boldsymbol{0}$. Similar to the so called 'leverage constant' in robust linear regression (Flores, 2015; Liu et al., 2018), we can define the block-wise 'incomplete' leverage constant $c_{\Omega_q}(\boldsymbol{W}_{end}) := \min_{|\bar{S}_\Omega|=q} \min_{\boldsymbol{z} \neq \boldsymbol{0}} \frac{\sum_{i \in S_\Omega} \|[\boldsymbol{W}_{end}\boldsymbol{z}]_i\|_2}{\sum_{i \in \Omega} \|[\boldsymbol{W}_{end}\boldsymbol{z}]_i\|_2}$, which is between 0 and 1, and is monotonic increasing when the number of corrupted blocks $q$ decreases. A sufficient condition for $\sum_{i \in S_\Omega} \|\boldsymbol{v}_i\|_2 - \sum_{i \in \bar{S}_\Omega} \|\boldsymbol{v}_i\|_2 > 0$ to hold is $c_{\Omega_q}(\boldsymbol{W}_{end}) > 0.5$. Lastly, it is very useful to note that Sharon et al. (2009) designed an algorithm to calculate the leverage constant, which is possible to be extended to our block-wise incomplete leverage constant here. Consider the toy example again where every agent holds exactly the same embedded features, and $q = 0.1 \times |\Omega|$, then $c_{\Omega_q}(\boldsymbol{W}_{end}) = 0.9$.

**Remark 3.** It is interesting to note that in the recent Generative model based compressive sensing work (Bora et al., 2017), they also impose conditions on the difference between any two vectors in the range of the decoder, though without missing blocks. But the conditions therein are quite different from ours due to the different purposes.

## 3.2 Noisy case

Now we analyze the 'noisy case' where the learned AutoEncoder can only approximately reconstruct the underlying uncorrupted feature vector $\boldsymbol{l}^*$, *i.e.*, $\sum_{i=1}^{M} \|[\mathcal{D}_\phi(\mathcal{E}_\psi(\boldsymbol{l}^*)) - \boldsymbol{l}^*]_i\|_2 \leq \delta$. Again, to help the reader understand step-by-step, our analyses start from the threat model B.

**Theorem 3. (Stable feature recovery under threat model B)** Assume the trained AutoEncoder satisfies $\sum_{i=1}^{M} \|[\mathcal{D}_\phi(\mathcal{E}_\psi(\boldsymbol{l}^*)) - \boldsymbol{l}^*]_i\|_2 \leq \delta$ for the underlying uncorrupted feature vector $\boldsymbol{l}^*$. Let $\hat{\boldsymbol{l}}$

be the solution of Eq. 6. Given the incomplete observations $\boldsymbol{h}_\Omega = \boldsymbol{l}_\Omega^*$, where $\Omega$ is the index set of observed agent blocks. For $\forall \boldsymbol{r}', \boldsymbol{r}''$ in the range of the decoder $\mathcal{D}_\phi(\cdot)$, define $\boldsymbol{v} = \boldsymbol{r}' - \boldsymbol{r}''$, if it holds that $\sum_{i\in\Omega} \|\boldsymbol{v}_i\|_2 > 2\delta$ for $\forall \|\boldsymbol{v}\|_2 > \Delta$, then $\|\mathcal{D}_\phi(\mathcal{E}_\psi(\hat{\boldsymbol{l}})) - \mathcal{D}_\phi(\mathcal{E}_\psi(\boldsymbol{l}^*))\|_2 \le \Delta$.

The proof can be found in Section A.5 of the supplemental material.

**Remark 4.** First, note that $\sum_{i=1}^M \|\boldsymbol{v}_i\|_2 \ge \sqrt{\sum_{i=1}^M \|\boldsymbol{v}_i\|_2^2} = \|\boldsymbol{v}\|_2$. When $\|\boldsymbol{v}\|_2 > \Delta$, we have $\sum_{i=1}^M \|\boldsymbol{v}_i\|_2 > \Delta$. Again, we prefer $\boldsymbol{v}$ to be not 'spiky', thus the condition $\sum_{i\in\Omega} \|\boldsymbol{v}_i\|_2 > 2\delta$ can be satisfied as long as $|\Omega|$ is large enough (assume $\Delta > 2\delta$).

Now we consider the most challenging case where we not only have missing features but also have corruptions in the observed features.

**Theorem 4. (Stable feature recovery under threat model C)** Assume the trained AutoEncoder satisfies $\sum_{i=1}^M \|[\mathcal{D}_\phi(\mathcal{E}_\psi(\boldsymbol{l}^*)) - \boldsymbol{l}^*]_i\|_2 \le \delta$ for the underlying uncorrupted feature vector $\boldsymbol{l}^*$. Let $\hat{\boldsymbol{l}}$ be the solution of Eq. 6. Given the incomplete observations $\boldsymbol{h}_\Omega = \boldsymbol{l}_\Omega^* + \boldsymbol{e}_\Omega^*$, where $\Omega$ is the index set of observed agent blocks. For $\forall \boldsymbol{r}', \boldsymbol{r}''$ in the range of the decoder $\mathcal{D}_\phi(\cdot)$ where $\boldsymbol{r}' \ne \boldsymbol{r}''$, define $\boldsymbol{v} = \boldsymbol{r}' - \boldsymbol{r}''$, if for any partition $\{S_\Omega, \bar{S}_\Omega\}$ of $\Omega$ with $|\bar{S}_\Omega| = q$, it holds that $\sum_{i\in S_\Omega} \|\boldsymbol{v}_i\|_2 - \sum_{i\in\bar{S}_\Omega} \|\boldsymbol{v}_i\|_2 > 2\delta$ for $\forall \|\boldsymbol{v}\|_2 > \Delta$, then $\|\mathcal{D}_\phi(\mathcal{E}_\psi(\hat{\boldsymbol{l}})) - \mathcal{D}_\phi(\mathcal{E}_\psi(\boldsymbol{l}^*))\|_2 \le \Delta$ as long as $\sum_{i\in\Omega} \mathbb{1}\{\boldsymbol{e}_i^* \ne \boldsymbol{0}\} \le q$.

The proof can be found in Section A.6 of the supplemental material.

**Robust prediction.** Recall that the server has learned a classifier $f_{\theta_0}$ during the training. We can further smooth this classifier $f_{\theta_0}$ to get $f_s$ (e.g., via Randomized Smoothing (Cohen et al., 2019; Lecuyer et al., 2019; Li et al., 2018)) such that $f_s(\mathcal{D}_\phi(\mathcal{E}_\psi(\hat{\boldsymbol{l}}))) = f_s(\mathcal{D}_\phi(\mathcal{E}_\psi(\hat{\boldsymbol{l}})) + \boldsymbol{v})$ for $\forall \|\boldsymbol{v}\|_2 \le \Delta$. As we have $\|\mathcal{D}_\phi(\mathcal{E}_\psi(\hat{\boldsymbol{l}})) - \mathcal{D}_\phi(\mathcal{E}_\psi(\boldsymbol{l}^*))\|_2 \le \Delta$, then it is guaranteed that $f_s(\mathcal{D}_\phi(\mathcal{E}_\psi(\hat{\boldsymbol{l}}))) = f_s(\mathcal{D}_\phi(\mathcal{E}_\psi(\boldsymbol{l}^*)))$.

**Remark 5.** The condition $\sum_{i\in S_\Omega} \|\boldsymbol{v}_i\|_2 - \sum_{i\in\bar{S}_\Omega} \|\boldsymbol{v}_i\|_2 > 2\delta$ required by Theorem 4 is stronger than the condition $\sum_{i\in\Omega} \|\boldsymbol{v}_i\|_2 > 2\delta$ required by Theorem 3. This is due to the presence of corruption in the observed blocks. As discussed in Remark 4, when $\|\boldsymbol{v}\|_2 > \Delta$, we have $\sum_{i=1}^M \|\boldsymbol{v}_i\|_2 > \Delta$. Again, we prefer $\boldsymbol{v}$ to be not 'spiky', thus $\sum_{i\in\Omega} \|\boldsymbol{v}_i\|_2$ can have a significant portion of mass from $\sum_{i=1}^M \|\boldsymbol{v}_i\|_2$ when the number of observed blocks $|\Omega|$ is large. Further, when the number of corrupted blocks $q = |\bar{S}_\Omega|$ is small enough, we can have $\sum_{i\in S_\Omega} \|\boldsymbol{v}_i\|_2 - \sum_{i\in\bar{S}_\Omega} \|\boldsymbol{v}_i\|_2 > 2\delta$. To draw the connections to the Range Space Property in robust linear regression, we continue on the discussions in Remark 2. Let's say $\sum_{i\in\Omega} \|\boldsymbol{v}_i\|_2 = \Delta'$, then a sufficient condition for $\sum_{i\in S_\Omega} \|\boldsymbol{v}_i\|_2 - \sum_{i\in\bar{S}_\Omega} \|\boldsymbol{v}_i\|_2 > 2\delta$ to hold is $[c_{\Omega_q}(\boldsymbol{W}_{end}) - (1 - c_{\Omega_q}(\boldsymbol{W}_{end}))] > 2\delta/\Delta'$. Intuitively, the smaller number $q$ of corrupted blocks and smaller $\delta$ (*i.e.*, better recovered feature subspace) as well as larger observed set $\Omega$ (then $\Delta'$ approaches $\Delta$), can lead to smaller recovery error bound $\Delta$.

## 4 Related work and discussion

**Handling missing data and outliers in IoT.** Sanyal & Zhang (2018) model the underlying uncorrupted IoT data as a low-dimensional linear subspace, *i.e.*, low-rank, and try to clean the corrupted sensor data by projecting them onto the estimated linear subspace. There is no guarantee of recovering the underlying uncorrupted sensor data. This approach is a linear version of the manifold projection method that we will discuss in detail.

Kekatos & Giannakis (2011) studied the robust sensing problem where each sensor $i$ has a linear measurement of the unknown parameter $\boldsymbol{x}$, *i.e.*, $\boldsymbol{b}_i = \boldsymbol{A}_i \boldsymbol{x}$, where both $\boldsymbol{b}_i$ and $\boldsymbol{A}_i$ are known and some sensors are additionally corrupted by large measurement errors, but without any missing values. They showed that it is possible to recover the true parameter $\boldsymbol{x}$ despite outlier corruptions. They also consider the scenario with dense inlier noise in the measurements, *i.e.*, where the measurements $\boldsymbol{b}_i$ only approximately lie on the range space of matrix $\boldsymbol{A}_i$, but their corresponding method does not have any recovery guarantee in such a noisy case.

There are also some works that study the scenario where the sensors observe different noisy versions of the *same* data vector, and they show it is possible to find a set of unattacked sensors based on the assumption that the fraction of attacked sensors is small, e.g., (Zhang et al., 2018) or utilizing additional knowledge that some sensors are not attacked (Wilson & Veeravalli, 2016). We refer interested readers to the survey paper (Zhang et al., 2018) for additional details on this line of work.

**Certified robustness for defending against adversarial attacks.** In the adversarial machine learning literature, many defense methods have been proposed to deal with adversarial perturbations. We refer the reader to Chakraborty et al. (2018) for a comprehensive survey. However, most of the existing defenses can only deal with *small* adversarial perturbations. Even worse, as new defenses are proposed, corresponding new attacks are also developed to adaptively attack them. Experimental demonstrations of a defense's efficacy based on currently existing attacks do not provide general proof of security. Therefore, certifiably robust defenses are of great interest. Randomized Smoothing (RS) is a promising technique (Cohen et al., 2019; Lecuyer et al., 2019; Li et al., 2018) which smoothes a base classifier to get a robust classifier, such that the prediction of this robust classifier on adversarially perturbed input is the same as its prediction on the original input when the $\ell_p$-norm of the adversarial perturbation is sufficiently bounded. Unfortunately, in our setting, the adversarial perturbation on compromised devices' embedded features can be very large. Further, RS does not have robustness guarantees when there are missing features.

Another promising defense is Randomized Ablation (Levine & Feizi, 2020), which was proposed to defend against sparse adversarial attacks where a small number of pixels in an image are changed to some arbitrary values in [0,1]. We extend this to the collaborative inference setting for comparison: the FC learns to make predictions based on randomly selected subsets of agent features during the training; and in the inference phase, FC randomly selects a subset of agents to make a prediction. The final prediction is by majority voting over the random trials. However, it is hard to design a base classifier that is able to train on randomly ablated input features[1]. Another limitation of this method is that its robust prediction (based on randomly ablated features) may not be the same as the attack-free vanilla setting (*i.e.*, the prediction of the vanilla classifier on the uncorrupted data). Further, to choose how many pixels to ablate, one needs knowledge of the number of corrupted pixels.

**Manifold Projection vs. Feature Purification.** In the centralized setting, several works (Meng & Chen, 2017; Ilyas et al., 2017; Lindqvist et al., 2018) proposed to project the adversarial perturbed image onto the manifold of the normal data to defend against adversarial attacks. A natural question is: why do we propose a feature purification method, instead of using manifold projection? First, the manifold projection approach can not guarantee the recovery of the underlying features. In fact, the resulting projected features can be arbitrarily far from the underlying true features if the adversarial perturbation can be arbitrarily large. To make this concrete, consider the linear manifold, *i.e.*, subspace. Suppose the underlying true feature vector $l$ is generated from the column space of a $d$ by $r$ tall matrix $U$. We observe the corrupted feature vector $h$, which can be decomposed as $l + e$, where $e$ models the adversarial perturbations. Projecting $h$ onto the column space of $U$ will get $P_U h = P_U l + P_U e = l + P_U e$, where $P_U = U(U^T U)^{-1} U^T$. So the recovery error is $P_U h - l = P_U e$. Apparently, if the adversarial perturbation vector $e$ is not orthogonal to this subspace (which is often the case), the recovery error would be proportional to the magnitude of the perturbation! It's hard to recover the underlying true feature even if we have a perfect subspace, and the problem looks even more challenging when the underlying manifold is non-linear. Fortunately, in our collaborative inference setting, there is important prior information about the pattern of the corruptions that we can leverage: the perturbation $e$ has a block-wise structure (each block corresponds to an agent), and the fraction of the perturbed blocks is small. Therefore, $e$ has a block-sparse structure. By exploring this fact, our proposed feature purification method can provably recover the underlying feature vector exactly or with bounded error, even if the underlying manifold is highly non-linear and the magnitude of the perturbation is arbitrarily large.

**Purifier Network vs. Feature Purification.** Besides the Manifold-Projection based approaches, one recent work (Naseer et al., 2020) tries to train a purifier neural network that maps any corrupted feature $h$ to the underlying true feature $l$, in the centralized setting. However, that method needs to generate the adversarial training examples based on the assumption that the magnitude of the adversarial perturbation $e$ is bounded by small $\epsilon$ (see Naseer et al. (2020, Eq.4)), which is different from our collaborative inference setting where the adversarial perturbation is unbounded. Further, in our collaborative inference setting, arbitrary $\alpha$ fraction of the blocks of feature $l$ can be perturbed by arbitrarily large values. Based on the high (combinatorial) cardinality of the input space, it is highly doubtful that one can learn a magic purifier network that is able to map a combinatorial number of different block-perturbation patterns (and with arbitrary unbounded perturbation values)

---

[1]For images, the authors designed a color channel approach to encode the absence of information (*i.e.*, NULL) at ablated pixels. Though a CNN-based classifier can be trained on such data, it still does not know such an encoding scheme of NULL.

to the underlying true feature $\boldsymbol{l}$. In sharp contrast, the proposed method can guarantee to recover the underlying true feature $\boldsymbol{l}$ exactly.

## 5 Empirical evaluation

We first describe the experimental setup and introduce a new attack in Section 5.1, then present experimental results in Section 5.2. More results can be found in Section B of the supplemental material.

### 5.1 Experimental setup

**Datasets.** We study the classification task on ExtraSensory (Vaizman et al., 2017) and NUS-WIDE (Chua et al., 2009) datasets. ExtraSensory contains the measurements from diverse sensors of smart phone and smart watch. We divide the sensors into ten agents, which correspond to the phone accelerometer, gyroscope, magnetometer, watch accelerometer, compass, location, audio, phone states (e.g., battery, wifi, ringing mode), environment sensing (e.g., light, air pressure, humidity, and temperature), and time-of-day (e.g., morning). There are naturally missing-feature problems, e.g., the user may not permit the use of microphone and/or location. The original purpose of this dataset is to recognize the behavioral context of the user (e.g., sitting, walking, and running) and recommend more appropriate music. We use the first 1721 samples from a user for training, and the rest 465 samples for testing, with the binary label 'sitting' or not.

In NUS-WIDE, each sample has 634 image features, 1000 text features, and 5 different labels, *i.e.*, 'buildings', 'grass', 'animal', 'water', 'person'. We split the features into four agents, where the 1st agent holds 360-d image features, 2nd agent holds the rest of the image features, and the remaining two agents each hold 500-d text features. We use 60000 samples for training, 1000 samples for testing targeted attacks, and 10000 samples for testing untargeted attacks.

**Attack Setup.** We perform the inference adversarial attacks, as well as their combination with the missing-feature attack (missing-feature attack alone is not very effective). For the adversarial attack, we test both targeted and untargeted attacks.

For the **targeted attack**, we adopt the commonly used Projected Gradient Descent (PGD) attack to generate adversarial perturbations on the embedded features held by malicious agents. However, since in this collaborative inference setting, the magnitude of the perturbation on the embedded features is allowed to be very large, we omit the projection step in PGD (or one can think of it as projecting onto a very large norm ball as the maximum perturbation constraint). In the ExtraSensory dataset, the target label is simply the opposite of the true label, since it is a binary classification. We set the PGD attack with a learning rate of 0.5 and 30 iterations so that it can successfully attack the unsecured model. While in the NUS-WIDE dataset, the target label is set to be 'grass'. We set the PGD attack with a learning rate of 0.1 and 50 iterations. Note that PGD attack is a white-box attack, where the attacker tries to find $\arg\min_{\boldsymbol{h}_{adv}} \ell(f_{\theta_0}(\{\boldsymbol{h}_{adv}, \boldsymbol{h}_{benign}\}), y^A)$ via using the gradient $d\ell(f_{\theta_0}(\{\boldsymbol{h}_{adv}, \boldsymbol{h}_{benign}\}), y^A)/d\boldsymbol{h}_{adv}$, where $\boldsymbol{h}_{adv}$ denotes the overall embedded features held by malicious agents and $y^A$ is the target label. In practice, the attacker may not easily obtain such gradient information, which usually needs to know the trained global model $f_{\theta_0}$ and $\boldsymbol{h}_{benign}$. Nevertheless, this attack serves as a smart targeted perturbation on the embedded features to test the robustness of the defense methods.

Motivated by the discussions in Section 4, we propose and test an **untargeted attack**, which is agnostic to the model, training data, label, as well as features held by other agents. We call it **distributed feature-flipping attack**. In this attack, the malicious agent(s) simply flips the sign of its embedded feature vector $\boldsymbol{l}_i$ and further amplifies its magnitude. Formally, during this inference phase untargeted attack, the malicious agent $i$ provides the embedded features $\boldsymbol{h}_i = -\text{Amplification} \times \boldsymbol{l}_i$ to the FC, instead of providing the true embedded features $\boldsymbol{l}_i$. This attack can be easily understood in binary classification with a linear classifier since flipping the sign of the whole feature vector will likely flip the sign of the predicted label as well. To illustrate the effectiveness of this attack against the manifold-projection based defenses, we can look it as adding the corruption vector $\boldsymbol{e}_i = -(1 + \text{Amplification}) \times \boldsymbol{l}_i$ to the true embedded feature vector $\boldsymbol{l}_i$ of $i$-th agent, which is malicious. So the overall perturbation vector $\boldsymbol{e}$ has many zero blocks corresponding to benign agents. Its non-zero blocks have values $\boldsymbol{e}_i = -(1 + \text{Amplification}) \times \boldsymbol{l}_i, \forall i$ corresponds to the malicious agents. So the inner product between perturbation vector $\boldsymbol{e}$ and true embedded features $\boldsymbol{l}$ is $-(1 + \text{Amplification}) \sum_{i \in \Omega_{adv}} \|\boldsymbol{l}_i\|_2^2$. Such perturbation vector $\boldsymbol{e}$ introduces correlations with

the underlying manifold of the uncorrupted embedded features. Therefore, projecting the corrupted embedded features $\boldsymbol{h}$ onto the manifold is hard to get rid of such corruptions. We will see such feature-flipping attack is very effective in attacking many existing defenses.

**Baseline methods.** We compare with 1) **Unsecured**: no defense. 2) **Manifold Projection**: project the corrupted embedded features $\boldsymbol{h}^{test}$ onto the learned manifold via Autoencoder, *i.e.*, $D_\phi(\mathcal{E}_\psi(\boldsymbol{h}^{test}))$. 3) **Randomized Smoothing** (Cohen et al., 2019), where randomly generated Gaussian noise is added to $\boldsymbol{h}^{test}$ for the FC to make the prediction, the final prediction is by majority voting over 1000 such random trials. 4) **Randomized Smoothing Block**: since only some block(s) of the embedded features are corrupted, we further compare with an *oracle* version of Randomized Smoothing, which only adds random Gaussian noise to the corrupted blocks of $\boldsymbol{h}^{test}$. 5) **Randomized Ablation** (Levine & Feizi, 2020) 6) **Adversarial Training (AT)** (Madry et al., 2018): standard adversarial training via using PGD attacks on $\boldsymbol{h}^{train}$ to generate adversarial examples, but without using projection step to limit the magnitude of adversarial perturbations. For defending against the targeted attack, it *knows the target label* and leverages that to generate targeted adversarial examples. For defending against untargeted attack, it uses untargeted PGD attack to generate adversarial examples. 7) **Block-wise Adversarial Training (BAT)**: since only some block(s) of $\boldsymbol{h}^{test}$ will be corrupted, we further propose and test a block-wise version of AT: for each training sample's embedded features, we randomly select a block and generate adversarial perturbations (via corresponding targeted or untargeted PGD attack) on that block for adversarial training. 8) **Oracle**: where there is no attack. The implementation details of the baselines and the proposed CoPur can be found in Section C of the supplemental material.

## 5.2 Empirical results

We report the Robust Accuracy of each defense method, *i.e.*, classification accuracy on the adversarial corrupted embedded features, against the distributed adversarial attack and its combination with the missing-feature attack. More experiments can be found in Section B of the supplemental material.

**A) Adversarial attack.** We first test the robustness of the defense methods under the targeted attack. Table 1 shows the Robust Accuracy of each method against targeted PGD attacks on NUS-WIDE and ExtraSensory datasets, where the last agent is malicious. The Manifold Projection method (which has no recovery guarantee) shows some resistance on the NUS-WIDE dataset but is not robust on the ExtraSensory dataset. The proposed method achieves the best Robust Accuracy on both datasets.

In Figure 2a, we report the Robust Accuracy of the defense methods under the untargeted feature-flipping attack on the ExtraSensory dataset, with Amplification ranging from [1, 3, 5, 10, 20], and the last three agents perform such attack. The Robust Accuracy of the compared methods drops significantly when the Amplification gets larger. The proposed method significantly outperforms the baselines by a large margin.

Table 1: Robust Accuracy (%) of each defense against targeted PGD attacks on NUS-WIDE and ExtraSensory datasets, where the last agent is malicious.

| | Unsecured | Manifold Proj | Rand Smooth | Rand Smooth Block | Rand Ablation | Adv Train | Block Adv Train | Proposed |
|---|---|---|---|---|---|---|---|---|
| NUS-WIDE | 4.5±0.3 | 81.3±1.2 | 41.8±6.3 | 48.7±10.1 | 65.3±5.3 | 8.0±1.9 | 7.3±1.2 | **83.7± 0.9** |
| ExtraSensory | 40.4± 16.9 | 53.6 ± 35.7 | 42.4 ± 17.9 | 42.0 ± 17.8 | 59.6± 27.5 | 19.8 ± 34.1 | 0.0± 0.0 | **79.0± 5.0** |

We are also interested in knowing the robustness limit of each defense when there are more and more agents performing adversarial attacks. Figure 3 shows the Robust Accuracy of each defense method w.r.t. the index set of agents that perform untargeted adversarial attack (Amplification=10) on ExtraSensory dataset. The Robustness Accuracy of the Adversarial Training method (green solid line) significantly drops when there are more than two malicious agents. The proposed method keeps very high Robust Accuracy even when agents 8-10 perform attack together. When agents 7-10 perform attack together, the Robust Accuracy of the proposed method apparently drops, but it is still significantly better than other baseline methods.

**B) Combined attack.** We now test the combined attack on ExtraSensory dataset, where agent 7 which holds audio MFCC features, performs missing-feature attack, and agent 10 performs untargeted (Figure 2b) or targeted (Table 2) adversarial attack. The proposed method significantly outperforms the baseline methods and maintains very high Robust Accuracy under both attacks.

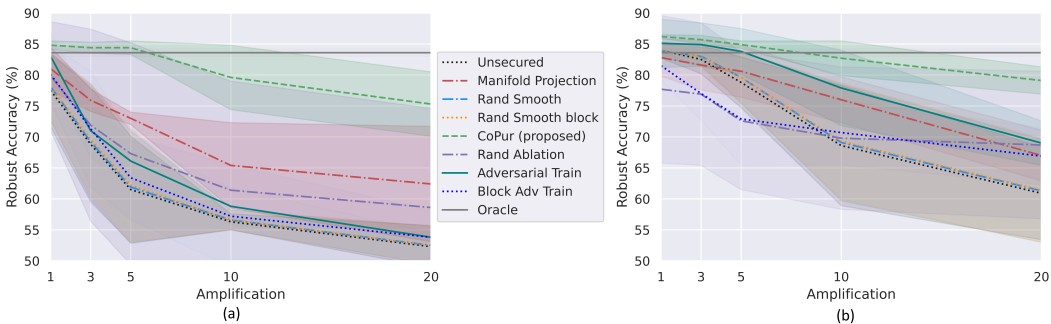

Figure 2: Robust Accuracy of each defense method under the untargeted adversarial attack and combined attacks on ExtraSensory dataset with 10 agents: (a) Agents 8,9,10 provide corrupted embedded features $h_{8,9,10} = -\text{Amplification} \times l_{8,9,10}$. (b) Agent 7 performs missing-feature attack, while agent 10 provides corrupted embedded features $h_{10} = -\text{Amplification} \times l_{10}$.

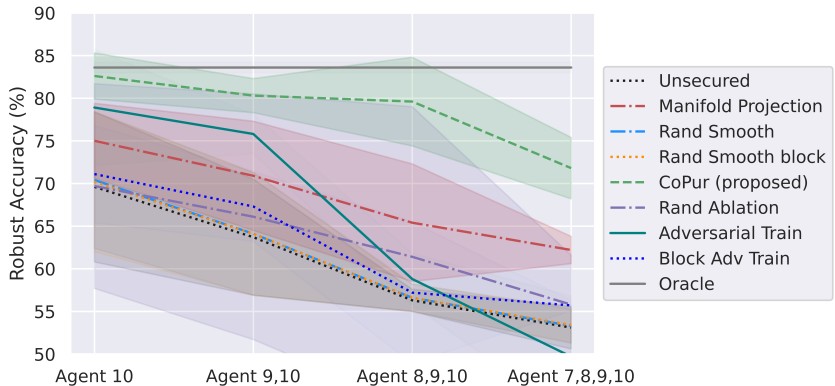

Figure 3: Robust Accuracy (%) of each defense method w.r.t. the index set of agents that perform untargeted adversarial attack (Amplification=10) on ExtraSensory dataset.

Table 2: Robust Accuracy (%) of each defense against combined targeted attacks (agent 7 performs missing-feature attack and agent 10 performs adversarial targeted attack) on ExtraSensory dataset.

| Unsecured | Manifold Proj | Rand Smooth | Rand Smooth Block | Rand Ablation | Adv Train | Block Adv Train | Proposed |
|---|---|---|---|---|---|---|---|
| 38.3± 15.7 | 52.6± 38.9 | 50.5± 1.3 | 51.0± 2.5 | 60.3± 28.1 | 24.6± 26.8 | 0.1± 0.1 | **79.1± 4.5** |

## 6  Conclusions

In this work, we proposed a novel feature purification based robust collaborative inference framework to defend against a variety of attacks during inference, with theoretical guarantees. The proposed non-linear robust decomposition method and its theoretical analyses may have much wider impacts. We further validate the robustness of the proposed framework through extensive experiments on ExtraSensory and NUS-WIDE datasets. Our future work includes extending Algorithm 1 of Sharon et al. (2009) to calculate the block-wise incomplete leverage constant introduced in Remark 2.

## Acknowledgments and Disclosure of Funding

This work is partially supported by NSF grant No.1910100, NSF CNS No.2046726, NSF III 2046795, IIS 1909577, CCF 1934986 and NIFA award 2020-67021-32799, a C3.ai DTI Award, a Jump Arches Award, and the Alfred P. Sloan Foundation. S.K. was supported by Google Inc. The authors thank the anonymous reviewers for their constructive suggestions. J.L. also thanks Dr. Pedro Cisneros-Velarde and Dr. Yu Ding for the helpful discussions regarding manifolds.

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
