# Supplemental Material for CoPur

## A  Additional theoretical results, definitions, and proofs

### A.1  Exact recovery when there is no attack

The reader may wonder how the proposed feature purification method behaves when there is no attack, i.e., $\boldsymbol{h} = \boldsymbol{l}^*$. The following theorem shows that we can recover $\boldsymbol{l}^*$ exactly even if it only approximately lies on the manifold.

**Theorem 5. (Exact feature recovery )** For the uncorrupted feature vector $\boldsymbol{h} = \boldsymbol{l}^*$, assume the learned AutoEncoder satisfies $\sum_{i=1}^{M} \|[\mathcal{D}_\phi(\mathcal{E}_\psi(\boldsymbol{l}^*)) - \boldsymbol{l}^*]_i\|_2 \leq \delta$. Let $\hat{\boldsymbol{l}}$ be the solution of Eq. 6. Then we have $\hat{\boldsymbol{l}} = \boldsymbol{l}^*$.

*Proof.* Since there is no attack, $\Omega = \{1, 2, ..., M\}$ in Eq. 6. It is straightforward to see $\hat{\boldsymbol{l}} = \boldsymbol{l}^*$ is the unique global minimum of Eq. 6, as it achieves the minimal zero value of the loss function.  □

### A.2  Block-wise incomplete leverage constant

In robust linear regression literature, for characterizing the robustness of the $\ell_1$ estimator, the following 'leverage constant' of a matrix $\boldsymbol{A}^{M \times D}$ is introduced in Flores (2015):

**Definition 1. (leverage constant)** Define for every $q \in \{1, ..., M\}$ the leverage constant $c_q$ of $\boldsymbol{A}^{M \times D}$ as

$$c_q(\boldsymbol{A}) := \min_{|\bar{S}|=q} \min_{\boldsymbol{z} \in \mathbb{R}^D \setminus \{\boldsymbol{0}\}} \frac{\sum_{i \in S} |\boldsymbol{a}_i^T \boldsymbol{z}|}{\sum_{i=1}^{M} |\boldsymbol{a}_i^T \boldsymbol{z}|},$$

where $\boldsymbol{a}_i^T$ denotes the $i$-th row of matrix $\boldsymbol{A}$, set $\bar{S} = \{1, ..., M\} \setminus S$.

The leverage constant $c_q$ is between 0 and 1, and is monotonic increasing when $q$ decreases.

In our analysis, we introduce the block-wise 'incomplete' leverage constant of matrix $\boldsymbol{W}$, whose rows are divided into $M$ blocks (each block corresponds to an agent), and we use $\Omega$ to denote the index set of observed blocks:

$$c_{\Omega_q}(\boldsymbol{W}) := \min_{|\bar{S}_\Omega|=q} \min_{\boldsymbol{z} \in \mathbb{R}^D \setminus \{\boldsymbol{0}\}} \frac{\sum_{i \in S_\Omega} \|[\boldsymbol{W}\boldsymbol{z}]_i\|_2}{\sum_{i \in \Omega} \|[\boldsymbol{W}\boldsymbol{z}]_i\|_2},$$

where $[\boldsymbol{W}\boldsymbol{z}]_i$ denotes $i$-th block of the vector $\boldsymbol{W}\boldsymbol{z}$, and $\bar{S}_\Omega = \Omega \setminus S_\Omega$. $c_{\Omega_q}(\boldsymbol{W})$ is between 0 and 1, and is monotonic increasing when the number of corrupted blocks $q$ decreases.

As mentioned in Remark 2, a sufficient condition for the condition $\sum_{i \in S_\Omega} \|\boldsymbol{v}_i\|_2 - \sum_{i \in \bar{S}_\Omega} \|\boldsymbol{v}_i\|_2 > 0$ that required by Theorem 2 to hold is $c_{\Omega_q}(\boldsymbol{W}_{end}) > 0.5$, where $\boldsymbol{W}_{end}$ denotes the weight matrix of the last layer of the AutoEncoder. Consider the toy example where every agent holds exactly the same embedded features, and $q = 0.1 \times |\Omega|$, then $c_{\Omega_q}(\boldsymbol{W}_{end}) = 0.9$.

### A.3  Proof of Theorem 1

*Proof.* Let $\boldsymbol{l}'$ be any feasible point of Eq. 5 (i.e., satisfies $\mathcal{D}_\phi(\mathcal{E}_\psi(\boldsymbol{l}')) = \boldsymbol{l}'$) that is different from $\boldsymbol{l}^*$. Let $\boldsymbol{v} = \mathcal{D}_\phi(\mathcal{E}_\psi(\boldsymbol{l}^*)) - \mathcal{D}_\phi(\mathcal{E}_\psi(\boldsymbol{l}')) = \boldsymbol{l}^* - \boldsymbol{l}'$. By the assumption of Theorem 1, we have $\boldsymbol{v}_\Omega \neq \boldsymbol{0}$. Then,

$$\sum_{i \in \Omega} \|[\boldsymbol{h} - \boldsymbol{l}']_i\|_2 \tag{7}$$

$$= \sum_{i \in \Omega} \|[\boldsymbol{h} - \mathcal{D}_\phi(\mathcal{E}_\psi(\boldsymbol{l}'))]_i\|_2 \tag{8}$$

$$= \sum_{i \in \Omega} \|[\boldsymbol{h} - \mathcal{D}_\phi(\mathcal{E}_\psi(\boldsymbol{l}^*)) + \boldsymbol{v}]_i\|_2 \tag{9}$$

$$= \sum_{i \in \Omega} \|[\boldsymbol{h} - \boldsymbol{l}^* + \boldsymbol{v}]_i\|_2 \tag{10}$$

$$= \sum_{i \in \Omega} \|\boldsymbol{v}_i\|_2 \tag{11}$$

$$> 0 \tag{12}$$

$$= \sum_{i \in \Omega} \|[\boldsymbol{h} - \boldsymbol{l}^*]_i\|_2, \tag{13}$$

Therefore, $\boldsymbol{l}^*$ is the unique global optimal solution of Eq. 5.

Similarly, let $\boldsymbol{l}'$ be any feasible point of Eq. 4 (i.e., satisfies $\mathcal{D}_\phi(\mathcal{E}_\psi(\boldsymbol{l}')) = \boldsymbol{l}'$) that is different from $\boldsymbol{l}^*$. Let $\boldsymbol{v} = \mathcal{D}_\phi(\mathcal{E}_\psi(\boldsymbol{l}^*)) - \mathcal{D}_\phi(\mathcal{E}_\psi(\boldsymbol{l}')) = \boldsymbol{l}^* - \boldsymbol{l}'$. By the assumption of Theorem 1, we have $\boldsymbol{v}_\Omega \neq \boldsymbol{0}$. Then,

$$\sum_{i \in \Omega} \mathbb{1}\{[\boldsymbol{h} - \boldsymbol{l}']_i \neq \boldsymbol{0}\} \tag{14}$$

$$= \sum_{i \in \Omega} \mathbb{1}\{[\boldsymbol{h} - \mathcal{D}_\phi(\mathcal{E}_\psi(\boldsymbol{l}'))]_i \neq \boldsymbol{0}\} \tag{15}$$

$$= \sum_{i \in \Omega} \mathbb{1}\{[\boldsymbol{h} - \mathcal{D}_\phi(\mathcal{E}_\psi(\boldsymbol{l}^*)) + \boldsymbol{v}]_i \neq \boldsymbol{0}\} \tag{16}$$

$$= \sum_{i \in \Omega} \mathbb{1}\{[\boldsymbol{h} - \boldsymbol{l}^* + \boldsymbol{v}]_i \neq \boldsymbol{0}\} \tag{17}$$

$$= \sum_{i \in \Omega} \mathbb{1}\{\boldsymbol{v}_i \neq \boldsymbol{0}\} \tag{18}$$

$$> 0 \tag{19}$$

$$= \sum_{i \in \Omega} \mathbb{1}\{[\boldsymbol{h} - \boldsymbol{l}^*]_i \neq \boldsymbol{0}\} \tag{20}$$

Therefore, $\boldsymbol{l}^*$ is the unique global optimal solution of Eq. 4. $\qquad\square$

### A.4 Proof of Theorem 2

*Proof.* A) Let $\boldsymbol{l}'$ be any feasible point of Eq. 5 (i.e., satisfies $\mathcal{D}_\phi(\mathcal{E}_\psi(\boldsymbol{l}')) = \boldsymbol{l}'$) that is different from $\boldsymbol{l}^*$. Let $\mathcal{D}_\phi(\mathcal{E}_\psi(\boldsymbol{l}')) = \mathcal{D}_\phi(\mathcal{E}_\psi(\boldsymbol{l}^*)) - \boldsymbol{v}$. Let $S_\Omega$ be any subset of $\Omega$ of size $|\Omega| - q$ such that $\boldsymbol{e}_i^* = \boldsymbol{0}, \forall i \in S_\Omega$. Denote the set difference $\Omega \setminus S_\Omega$ as $\bar{S}_\Omega$.

$$\sum_{i \in \Omega} \|[\boldsymbol{h} - \boldsymbol{l}']_i\|_2 \tag{21}$$

$$= \sum_{i \in \Omega} \|[\boldsymbol{h} - \mathcal{D}_\phi(\mathcal{E}_\psi(\boldsymbol{l}'))]_i\|_2 \tag{22}$$

$$= \sum_{i \in \Omega} \|[\boldsymbol{h} - \mathcal{D}_\phi(\mathcal{E}_\psi(\boldsymbol{l}^*)) + \boldsymbol{v}]_i\|_2 \tag{23}$$

$$= \sum_{i \in \Omega} \|[\boldsymbol{h} - \mathcal{D}_\phi(\mathcal{E}_\psi(\boldsymbol{l}^*))]_i + \boldsymbol{v}_i\|_2 \tag{24}$$

$$= \sum_{i \in S_\Omega} \|[\boldsymbol{h} - \mathcal{D}_\phi(\mathcal{E}_\psi(\boldsymbol{l}^*))]_i + \boldsymbol{v}_i\|_2 + \sum_{i \in \bar{S}_\Omega} \|[\boldsymbol{h} - \mathcal{D}_\phi(\mathcal{E}_\psi(\boldsymbol{l}^*))]_i + \boldsymbol{v}_i\|_2 \tag{25}$$

$$= \sum_{i \in S_\Omega} \|[\boldsymbol{h} - \boldsymbol{l}^*]_i + \boldsymbol{v}_i\|_2 + \sum_{i \in \bar{S}_\Omega} \|[\boldsymbol{h} - \mathcal{D}_\phi(\mathcal{E}_\psi(\boldsymbol{l}^*))]_i + \boldsymbol{v}_i\|_2 \tag{26}$$

$$= \sum_{i \in S_\Omega} \|\boldsymbol{v}_i\|_2 + \sum_{i \in \bar{S}_\Omega} \|[\boldsymbol{h} - \mathcal{D}_\phi(\mathcal{E}_\psi(\boldsymbol{l}^*))]_i + \boldsymbol{v}_i\|_2 \tag{27}$$

$$\geq \sum_{i \in S_\Omega} \|\boldsymbol{v}_i\|_2 + \sum_{i \in \bar{S}_\Omega} \|[\boldsymbol{h} - \mathcal{D}_\phi(\mathcal{E}_\psi(\boldsymbol{l}^*))]_i\|_2 - \sum_{i \in \bar{S}_\Omega} \|\boldsymbol{v}_i\|_2 \tag{28}$$

$$> \sum_{i \in \bar{S}_\Omega} \|[\boldsymbol{h} - \mathcal{D}_\phi(\mathcal{E}_\psi(\boldsymbol{l}^*))]_i\|_2 \tag{29}$$

$$= \sum_{i \in \bar{S}_\Omega} \|[\boldsymbol{h} - \boldsymbol{l}^*]_i\|_2 \tag{30}$$

$$= \sum_{i \in \Omega} \|[\boldsymbol{h} - \boldsymbol{l}^*]_i\|_2 \tag{31}$$

Therefore, $\boldsymbol{l}^*$ is the unique global optimal solution of Eq. 5.

B) First, note that $\boldsymbol{h}_\Omega = \boldsymbol{l}^*_\Omega + \boldsymbol{e}^*_\Omega$ and $\sum_{i \in \Omega} \mathbb{1}\{\boldsymbol{e}^*_i \neq \boldsymbol{0}\} \leq q$. Suppose $\boldsymbol{l}'$ is a global optimal solution of Eq. 4 that is different from $\boldsymbol{l}^*$. Let $\boldsymbol{h}_\Omega = \boldsymbol{l}'_\Omega + \boldsymbol{e}'_\Omega$, then we have

$$\sum_{i \in \Omega} \mathbb{1}\{\boldsymbol{e}'_i \neq \boldsymbol{0}\} \leq \sum_{i \in \Omega} \mathbb{1}\{\boldsymbol{e}^*_i \neq \boldsymbol{0}\} \leq q \tag{32}$$

and $\boldsymbol{l}'_\Omega + \boldsymbol{e}'_\Omega = \boldsymbol{l}^*_\Omega + \boldsymbol{e}^*_\Omega$.

From Eq. 32 we know that

$$\sum_{i \in \Omega} \mathbb{1}\{[\boldsymbol{l}^* - \boldsymbol{l}']_i = \boldsymbol{0}\} \geq |\Omega| - \sum_{i \in \Omega} \mathbb{1}\{\boldsymbol{e}^*_i \neq \boldsymbol{0}\} - \sum_{i \in \Omega} \mathbb{1}\{\boldsymbol{e}'_i \neq \boldsymbol{0}\} \tag{33}$$

$$\geq |\Omega| - q - q = |\Omega| - 2q \tag{34}$$

Let $\boldsymbol{v} = \mathcal{D}_\phi(\mathcal{E}_\psi(\boldsymbol{l}^*)) - \mathcal{D}_\phi(\mathcal{E}_\psi(\boldsymbol{l}')) = \boldsymbol{l}^* - \boldsymbol{l}'$, so we have $\sum_{i \in \Omega} \mathbb{1}\{\boldsymbol{v}_i = \boldsymbol{0}\} \geq |\Omega| - 2q$. Now split $\Omega$ into 3 disjoint sets $\{\Omega_0, \Omega_1, \Omega_2\}$, where $\Omega_0$ is any subset of $\Omega$ with size $|\Omega| - 2q$ such that $\boldsymbol{v}_{\Omega_0} = \boldsymbol{0}$, and $|\Omega_1| = |\Omega_2| = q$. Since $|\Omega_0 \cup \Omega_1| = |\Omega| - q$, by our assumption, we have $\sum_{i \in \Omega_0 \cup \Omega_1} \|\boldsymbol{v}_i\|_2 > \sum_{i \in \Omega_2} \|\boldsymbol{v}_i\|_2$. Since $|\Omega_0 \cup \Omega_2| = |\Omega| - q$, by our assumption, we have $\sum_{i \in \Omega_0 \cup \Omega_2} \|\boldsymbol{v}_i\|_2 > \sum_{i \in \Omega_1} \|\boldsymbol{v}_i\|_2$. However, this leads to a contradiction since $\sum_{i \in \Omega_0} \|\boldsymbol{v}_i\|_2 = 0$. $\qquad \square$

## A.5  Proof of Theorem 3

*Proof.* Consider any feasible point $\boldsymbol{l}'$ of Eq. 6 such that $\mathcal{D}_\phi(\mathcal{E}_\psi(\boldsymbol{l}')) \neq \mathcal{D}_\phi(\mathcal{E}_\psi(\boldsymbol{l}^*))$, and let $\boldsymbol{v} = \mathcal{D}_\phi(\mathcal{E}_\psi(\boldsymbol{l}^*)) - \mathcal{D}_\phi(\mathcal{E}_\psi(\boldsymbol{l}'))$.

$$\sum_{i \in \Omega} \|[\boldsymbol{h} - \boldsymbol{l}']_i\|_2 \tag{35}$$

$$= \sum_{i \in \Omega} \|[\boldsymbol{h} - \mathcal{D}_\phi(\mathcal{E}_\psi(\boldsymbol{l}')) + \mathcal{D}_\phi(\mathcal{E}_\psi(\boldsymbol{l}')) - \boldsymbol{l}']_i\|_2 \tag{36}$$

$$\geq \sum_{i \in \Omega} \|[\boldsymbol{h} - \mathcal{D}_\phi(\mathcal{E}_\psi(\boldsymbol{l}'))]_i\|_2 - \sum_{i \in \Omega} \|[\mathcal{D}_\phi(\mathcal{E}_\psi(\boldsymbol{l}')) - \boldsymbol{l}']_i\|_2 \tag{37}$$

$$\geq \sum_{i \in \Omega} \|[\boldsymbol{h} - \mathcal{D}_\phi(\mathcal{E}_\psi(\boldsymbol{l}'))]_i\|_2 - \delta \tag{38}$$

$$= \sum_{i \in \Omega} \|[\boldsymbol{h} - \mathcal{D}_\phi(\mathcal{E}_\psi(\boldsymbol{l}^*))]_i + \boldsymbol{v}_i\|_2 - \delta \tag{39}$$

$$= \sum_{i \in \Omega} \|[\boldsymbol{h} - \boldsymbol{l}^* + \boldsymbol{l}^* - \mathcal{D}_\phi(\mathcal{E}_\psi(\boldsymbol{l}^*))]_i + \boldsymbol{v}_i\|_2 - \delta \tag{40}$$

$$= \sum_{i \in \Omega} \|[\boldsymbol{l}^* - \mathcal{D}_\phi(\mathcal{E}_\psi(\boldsymbol{l}^*))]_i + \boldsymbol{v}_i\|_2 - \delta \tag{41}$$

$$\geq \sum_{i \in \Omega} \|\boldsymbol{v}_i\|_2 - \sum_{i \in \Omega} \|[\boldsymbol{l}^* - \mathcal{D}_\phi(\mathcal{E}_\psi(\boldsymbol{l}^*))]_i\|_2 - \delta \tag{42}$$

$$\geq \sum_{i \in \Omega} \|\boldsymbol{v}_i\|_2 - 2\delta \tag{43}$$

$$= \sum_{i \in \Omega} \|[\boldsymbol{h} - \boldsymbol{l}^*]_i\|_2 + \sum_{i \in \Omega} \|\boldsymbol{v}_i\|_2 - 2\delta \tag{44}$$

where the last equality follows from the fact that $\sum_{i \in \Omega} \|[\boldsymbol{h} - \boldsymbol{l}^*]_i\|_2 = 0$. If $\|\boldsymbol{v}\|_2 > \Delta$, the condition $\sum_{i \in \Omega} \|\boldsymbol{v}_i\|_2 > 2\delta$ means $\sum_{i \in \Omega} \|\boldsymbol{v}_i\|_2 - 2\delta > 0$, and from Eq. 44 we have $\sum_{i \in \Omega} \|[\boldsymbol{h} - \boldsymbol{l}']_i\|_2 > \sum_{i \in \Omega} \|[\boldsymbol{h} - \boldsymbol{l}^*]_i\|_2$. Then, the optimal solution must satisfy $\|\boldsymbol{v}\|_2 \leq \Delta$, which completes the proof. $\qquad\square$

## A.6   Proof of Theorem 4

*Proof.* Consider any feasible point $\boldsymbol{l}'$ of Eq. 6 such that $\mathcal{D}_\phi(\mathcal{E}_\psi(\boldsymbol{l}')) \neq \mathcal{D}_\phi(\mathcal{E}_\psi(\boldsymbol{l}^*))$, and let $\boldsymbol{v} = \mathcal{D}_\phi(\mathcal{E}_\psi(\boldsymbol{l}^*)) - \mathcal{D}_\phi(\mathcal{E}_\psi(\boldsymbol{l}'))$. Let $S_\Omega$ be any subset of $\Omega$ of size $|\Omega| - q$ such that $\boldsymbol{e}_i^* = \boldsymbol{0}, \forall i \in S_\Omega$. Denote the set difference $\Omega \setminus S_\Omega$ as $\bar{S}_\Omega$.

$$\sum_{i \in \Omega} \|[\boldsymbol{h} - \boldsymbol{l}']_i\|_2 \tag{45}$$

$$= \sum_{i \in \Omega} \|[\boldsymbol{h} - \mathcal{D}_\phi(\mathcal{E}_\psi(\boldsymbol{l}')) + \mathcal{D}_\phi(\mathcal{E}_\psi(\boldsymbol{l}')) - \boldsymbol{l}']_i\|_2 \tag{46}$$

$$\geq \sum_{i \in \Omega} \|[\boldsymbol{h} - \mathcal{D}_\phi(\mathcal{E}_\psi(\boldsymbol{l}'))]_i\|_2 - \sum_{i \in \Omega} \|[\mathcal{D}_\phi(\mathcal{E}_\psi(\boldsymbol{l}')) - \boldsymbol{l}']_i\|_2 \tag{47}$$

$$\geq \sum_{i \in \Omega} \|[\boldsymbol{h} - \mathcal{D}_\phi(\mathcal{E}_\psi(\boldsymbol{l}'))]_i\|_2 - \delta \tag{48}$$

$$= \sum_{i \in \Omega} \|[\boldsymbol{h} - \mathcal{D}_\phi(\mathcal{E}_\psi(\boldsymbol{l}^*))]_i + \boldsymbol{v}_i\|_2 - \delta \tag{49}$$

$$= \sum_{i \in \Omega} \|[\boldsymbol{h} - \boldsymbol{l}^* + \boldsymbol{l}^* - \mathcal{D}_\phi(\mathcal{E}_\psi(\boldsymbol{l}^*))]_i + \boldsymbol{v}_i\|_2 - \delta \tag{50}$$

$$= \sum_{i \in S_\Omega} \|[\boldsymbol{h} - \boldsymbol{l}^* + \boldsymbol{l}^* - \mathcal{D}_\phi(\mathcal{E}_\psi(\boldsymbol{l}^*))]_i + \boldsymbol{v}_i\|_2 \tag{51}$$

$$+ \sum_{i \in \bar{S}_\Omega} \|[\boldsymbol{h} - \boldsymbol{l}^* + \boldsymbol{l}^* - \mathcal{D}_\phi(\mathcal{E}_\psi(\boldsymbol{l}^*))]_i + \boldsymbol{v}_i\|_2 - \delta \tag{52}$$

$$= \sum_{i \in S_\Omega} \|[\boldsymbol{l}^* - \mathcal{D}_\phi(\mathcal{E}_\psi(\boldsymbol{l}^*))]_i + \boldsymbol{v}_i\|_2 \tag{53}$$

$$+ \sum_{i \in \bar{S}_\Omega} \|[\boldsymbol{h} - \boldsymbol{l}^* + \boldsymbol{l}^* - \mathcal{D}_\phi(\mathcal{E}_\psi(\boldsymbol{l}^*))]_i + \boldsymbol{v}_i\|_2 - \delta \tag{54}$$

$$\geq \sum_{i \in S_\Omega} \|\boldsymbol{v}_i\|_2 - \sum_{i \in S_\Omega} \|[\boldsymbol{l}^* - \mathcal{D}_\phi(\mathcal{E}_\psi(\boldsymbol{l}^*))]_i\|_2 + \sum_{i \in \bar{S}_\Omega} \|[\boldsymbol{h} - \boldsymbol{l}^*]_i\|_2$$
$$- \sum_{i \in \bar{S}_\Omega} \|[\boldsymbol{l}^* - \mathcal{D}_\phi(\mathcal{E}_\psi(\boldsymbol{l}^*))]_i\|_2 - \sum_{i \in \bar{S}_\Omega} \|\boldsymbol{v}_i\|_2 - \delta \tag{55}$$

$$= \sum_{i \in S_\Omega} \|\boldsymbol{v}_i\|_2 - \sum_{i \in \Omega} \|[\boldsymbol{l}^* - \mathcal{D}_\phi(\mathcal{E}_\psi(\boldsymbol{l}^*))]_i\|_2 + \sum_{i \in \bar{S}_\Omega} \|[\boldsymbol{h} - \boldsymbol{l}^*]_i\|_2 - \sum_{i \in \bar{S}_\Omega} \|\boldsymbol{v}_i\|_2 - \delta \tag{56}$$

$$\geq \sum_{i \in S_\Omega} \|\boldsymbol{v}_i\|_2 - \delta + \sum_{i \in \bar{S}_\Omega} \|[\boldsymbol{h} - \boldsymbol{l}^*]_i\|_2 - \sum_{i \in \bar{S}_\Omega} \|\boldsymbol{v}_i\|_2 - \delta \tag{57}$$

$$= \sum_{i \in S_\Omega} \|\boldsymbol{v}_i\|_2 - \delta + \sum_{i \in \Omega} \|[\boldsymbol{h} - \boldsymbol{l}^*]_i\|_2 - \sum_{i \in \bar{S}_\Omega} \|\boldsymbol{v}_i\|_2 - \delta \tag{58}$$

$$= \sum_{i \in \Omega} \|[\boldsymbol{h} - \boldsymbol{l}^*]_i\|_2 + \sum_{i \in S_\Omega} \|\boldsymbol{v}_i\|_2 - \sum_{i \in \bar{S}_\Omega} \|\boldsymbol{v}_i\|_2 - 2\delta \tag{59}$$

If $\|\boldsymbol{v}\|_2 > \Delta$, the condition $\sum_{i \in S_\Omega} \|\boldsymbol{v}_i\|_2 - \sum_{i \in \bar{S}_\Omega} \|\boldsymbol{v}_i\|_2 > 2\delta$ means $\sum_{i \in S_\Omega} \|\boldsymbol{v}_i\|_2 - \sum_{i \in \bar{S}_\Omega} \|\boldsymbol{v}_i\|_2 - 2\delta > 0$, and from Eq. 59 we have $\sum_{i \in \Omega} \|[\boldsymbol{h} - \boldsymbol{l}']_i\|_2 > \sum_{i \in \Omega} \|[\boldsymbol{h} - \boldsymbol{l}^*]_i\|_2$. Then, the optimal solution must satisfy $\|\boldsymbol{v}\|_2 \leq \Delta$, which completes the proof.

$\square$

# B  Ablation studies

We first do an ablation study by comparing the recovered embedded features with the underlying uncorrupted embedded features on the NUS-WIDE dataset. Only the Manifold Projection method and the proposed method can reconstruct embedded features. Figure 4-5 show the reconstruction error of the embedded features on first 1000 testing instances by Manifold Projection method (i.e., $\|\mathcal{D}_\phi(\mathcal{E}_\psi(\boldsymbol{h})) - \mathcal{D}_\phi(\mathcal{E}_\psi(\boldsymbol{l}^*))\|_2$) and by the proposed feature purification method (i.e., $\|\mathcal{D}_\phi(\mathcal{E}_\psi(\hat{\boldsymbol{l}})) - \mathcal{D}_\phi(\mathcal{E}_\psi(\boldsymbol{l}^*))\|_2$), where $\boldsymbol{l}^*$ is the underlying uncorrupted embedded features. Apparently, the proposed feature purification method has a much smaller reconstruction error on almost every testing instance than the Manifold Projection method.

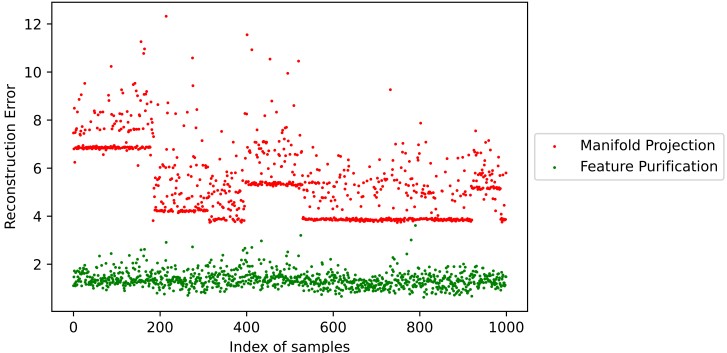

Figure 4: Reconstruction error of the embedded features by Manifold Projection (i.e., $\|\mathcal{D}_\phi(\mathcal{E}_\psi(\boldsymbol{h})) - \mathcal{D}_\phi(\mathcal{E}_\psi(\boldsymbol{l}^*))\|_2$, in red) and Feature Purification (i.e., $\|\mathcal{D}_\phi(\mathcal{E}_\psi(\hat{\boldsymbol{l}})) - \mathcal{D}_\phi(\mathcal{E}_\psi(\boldsymbol{l}^*))\|_2$, in green) on first 1000 testing samples of NUS-WIDE dataset, under the untargeted adversarial attack by agent 4 with Ampilification=10.

We then do some ablation studies by comparing the adversarial attack with the combined attack on NUS-WIDE and ExtraSensory datasets. In Table 3, the first row shows the Robust Accuracy of each defense against the targeted attack by agent 4 on NUS-WIDE dataset, while the second row shows the Robust Accuracy of each defense against the combined targeted attack (agent 4 performs targeted attack while agent 1 performs missing-feature attack). We can see that the Robust Accuracies of most methods drop when agent 1 additionally performs the missing-feature attack.

Table 3: Robust Accuracy (%) of each defense against targeted attack (by agent 4, first row) and the combined targeted attack (in addition, agent 1 performs missing-feature attack, second row) on NUS-WIDE dataset.

| | Unsecured | Manifold Proj | Rand Smooth | Rand Smooth Block | Rand Ablation | Adv Train | Block Adv Train | Proposed |
|---|---|---|---|---|---|---|---|---|
| #4 target | 4.5±0.3 | 81.3±1.2 | 41.8±6.3 | 48.7±10.1 | 65.3±5.3 | 8.0±1.9 | 7.3±1.2 | **83.7±0.9** |
| #1 miss, #4 target | 4.2± 0.4 | 70.7 ± 2.5 | 5.1 ± 1.1 | 44.4 ± 9.3 | 60.9± 1.3 | 4.6 ± 0.3 | 4.3± 0.3 | **71.8± 2.0** |

In Table 4, the first row shows the Robust Accuracy of each defense against the untargeted attack (with Amplification=10) by agent 4 on NUS-WIDE dataset, while the second row shows the Robust Accuracy of each defense against the combined untargeted attack (agent 4 performs such untargeted attack while agent 1 performs missing-feature attack). Again, the Robust Accuracies of most methods drop when agent 1 additionally performs the missing-feature attack. The Manifold Projection method almost fails under this combined attack. The proposed method is still resistant to such combined attack.

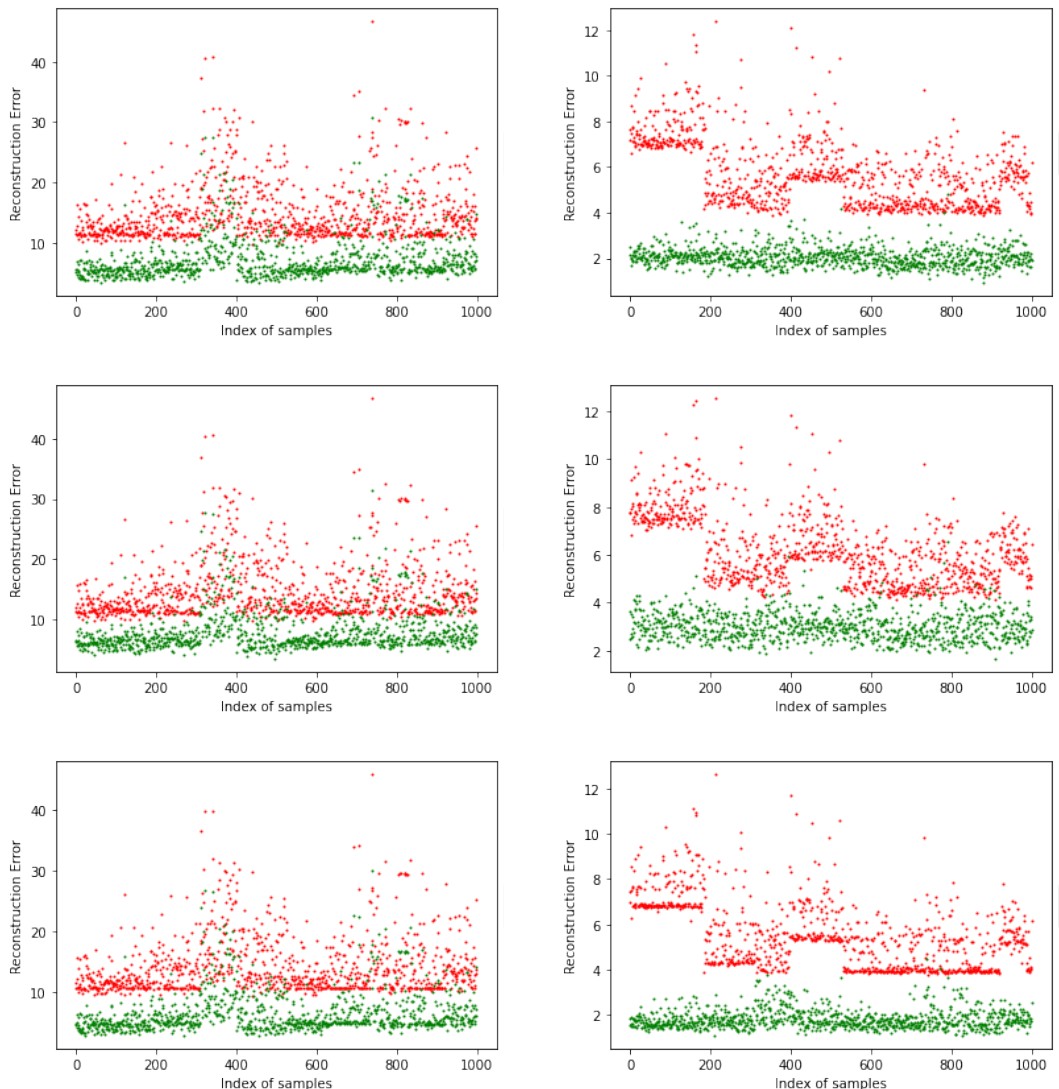

Figure 5: Reconstruction error of the embedded features by Manifold Projection (i.e., $\|\mathcal{D}_\phi(\mathcal{E}_\psi(\boldsymbol{h})) - \mathcal{D}_\phi(\mathcal{E}_\psi(\boldsymbol{l}^*))\|_2$, in red) and Feature Purification (i.e., $\|\mathcal{D}_\phi(\mathcal{E}_\psi(\hat{\boldsymbol{l}})) - \mathcal{D}_\phi(\mathcal{E}_\psi(\boldsymbol{l}^*))\|_2$, in green) on first 1000 testing samples of NUS-WIDE dataset, under the combined attacks with Ampilification=10: (a) Agent 1 performs missing-feature attack, Agent 3 performs adversarial untargeted attack; (b) Agent 1 performs missing-feature attack, Agent 4 performs adversarial untargeted attack; (c) Agent 2 performs missing-feature attack, Agent 3 performs adversarial untargeted attack; (d) Agent 2 performs missing-feature attack, Agent 4 performs adversarial untargeted attack; (e) Agent 4 performs missing-feature attack, Agent 3 performs adversarial untargeted attack; (f) Agent 3 performs missing-feature attack, Agent 4 performs adversarial untargeted attack.

Table 4: Robust Accuracy (%) of each defense against untargeted attack (agents 4 provides corrupted embedded features $h_4 = -10 \times l_4$, first row) and the combined untargeted attack (in addition, agent 1 performs missing-feature attack, second row) on NUS-WIDE dataset.

|  | Unsecured | Manifold Proj | Rand Smooth | Rand Smooth Block | Rand Ablation | Adv Train | Block Adv Train | Proposed |
|---|---|---|---|---|---|---|---|---|
| #4 untarget | 68.4± 17.5 | 64.3 ± 11.1 | 68.8 ± 16.9 | 70.8 ± 13.4 | 77.2± 2.1 | 80.8 ± 2.5 | 80.8± 2.5 | **82.1± 0.7** |
| #1 miss, #4 untarget | 61.7±15.6 | 52.1±16.7 | 62.8±16.4 | 62.3±12.2 | 73.6±4.6 | 69.2±5.6 | 69.2±5.6 | **74.1±2.3** |

Table 5: Robust Accuracy (%) of each defense against targeted attack and the combined targeted attack on ExtraSensory dataset. First row: agent 10 performs targeted attack; Second row: agent 10 performs targeted attack while agent 7 performs missing-feature attack; Third row: agent 10 performs targeted attack while agents 7&8 perform missing-feature attack; Last row: agent 10 performs targeted attack while agents 1&7 perform missing-feature attack.

|  | Unsecured | Manifold Proj | Rand Smooth | Rand Smooth Block | Rand Ablation | Adv Train | Block Adv Train | Proposed |
|---|---|---|---|---|---|---|---|---|
| #10 target | 40.4± 16.9 | 53.6 ± 35.7 | 42.4 ± 17.9 | 42.0 ± 17.8 | 59.6± 27.5 | 19.8 ± 34.1 | 0.0± 0.0 | **79.0± 5.0** |
| #7 miss, #10 target | 38.3± 15.7 | 52.6± 38.9 | 50.5± 1.3 | 51.0± 2.5 | 60.3± 28.1 | 24.6± 26.8 | 0.1± 0.1 | **79.1± 4.5** |
| #7,8 miss, #10 target | 36.0 ± 17.7 | 75.2 ± 7.6 | 38.4 ± 19.0 | 37.3± 18.9 | 59.7± 29.4 | 35.3± 30.6 | 0.0± 0.0 | **81.1 ± 2.6** |
| #1,7 miss, #10 target | 25.2 ± 17.3 | 65.6 ± 6.6 | 26.0 ± 18.3 | 25.9± 17.9 | 61.0± 28.6 | 10.7± 18.6 | 0.0± 0.0 | **71.6 ± 9.0** |

In Table 5, the first row shows the Robust Accuracy of each defense against the targeted attack by agent 10 on ExtraSensory dataset. The second row shows the Robust Accuracy when agent 10 performs targeted attack while agent 7, which holds the audio features, performs missing-feature attack; The third row shows the Robust Accuracy when agent 10 performs targeted attack, while agents 7&8 perform missing-feature attack; The last row shows the Robust Accuracy when agent 10 performs targeted attack while agents 1&7 perform missing-feature attack. The Robust Accuracy of the Unsecured method significantly drops when the adversarial attack is combined with more agents that perform missing-feature attack. Adversarial Training and Randomized Smoothing type methods almost fail under targeted attacks. Though the Robust Accuracy of the proposed method drops a bit when agents 1 & 7 additionally perform missing-feature attack, it still significantly outperforms other baseline defense methods.

In Table 6, the first row shows the Robust Accuracy of each defense against the untargeted attack by agent 10 on ExtraSensory dataset, where agent 10 provides corrupted embedded features $h_{10} = -10 \times l_{10}$. The second row shows the Robust Accuracy when agent 10 performs such untargeted attack while agent 7, which holds the audio features, performs missing-feature attack; The third row shows the Robust Accuracy when agent 10 performs untargeted attack while agents 7&8 perform missing-feature attack; The last row shows the Robust Accuracy when agent 10 performs untargeted attack while agents 1&7 perform missing-feature attack. The Robust Accuracy of the Unsecured method drops when the adversarial attack is combined with more agents that perform missing-feature attack. The proposed method keeps a very high Robust Accuracy, e.g., its Robust Accuracy in the last row is even higher than the baselines' Robust Accuracy in the first row.

Table 6: Robust Accuracy (%) of each defense against untargeted attack and the combined untargeted attack on ExtraSensory dataset. First row: agent 10 performs untargeted attack (by providing corrupted embedded features $h_{10} = -10 \times l_{10}$); Second row: agent 10 performs untargeted attack while agent 7 performs missing-feature attack; Third row: agent 10 performs untargeted attack while agents 7&8 perform missing-feature attack; Last row: agent 10 performs untargeted attack while agents 1&7 perform missing-feature attack.

|  | Unsecured | Manifold Proj | Rand Smooth | Rand Smooth Block | Rand Ablation | Adv Train | Block Adv Train | Proposed |
|---|---|---|---|---|---|---|---|---|
| #10 untarget | 69.6± 8.8 | 75.0 ± 4.4 | 70.4 ± 8.0 | 70.2 ± 8.2 | 69.7± 12.0 | 78.9 ± 6.8 | 71.1± 5.7 | **82.6± 2.7** |
| #7 miss, #10 untarget | 68.6± 9.7 | 76.0± 3.7 | 69.1± 9.4 | 69.3± 9.4 | 69.8± 11.5 | 77.9± 6.1 | 70.7± 5.2 | **82.7± 2.8** |
| #7,8 miss, #10 untarget | 68.1 ± 8.3 | 75.1 ± 5.1 | 68.9 ± 8.0 | 68.9± 7.9 | 75.3± 11.4 | 75.1± 10.4 | 64.2± 15.0 | **81.9 ± 2.7** |
| #1,7 miss, #10 untarget | 60.1 ± 6.2 | 68.8 ± 8.2 | 60.4 ± 6.7 | 60.4± 6.7 | 69.8± 11.5 | 68.7± 8.7 | 60.7± 13.7 | **80.1 ± 4.0** |

We further do some ablation studies by testing the Robust Accuracy of each defense when there are only missing-feature attacks, the results are shown in Table 7. The first row shows the Robust Accuracy when only agent 7 (which holds audio features) performs missing-feature attack; The second row shows the Robust Accuracy when agent 7&8 perform missing-feature attack; The last row shows the Robust Accuracy when agent 1&7 perform missing-feature attack. Comparing the last two rows, agent 1 seems to be more effective in performing missing-feature attack than agent 8. This makes sense since agent 1 corresponds to phone accelerometer, which is more useful than agent 8 (which holds phone state features like battery level) in predicting whether the user is sitting or not. Comparing with Tables 5-6, we can see that though missing-feature attack alone seems not very effective, by combining with adversarial targeted attack, the Robust Accuracy of many defenses further drop significantly.

Table 7: Robust Accuracy (%) of each defense under the missing-feature attacks by different agents on ExtraSensory dataset.

| | Unsecured | Manifold Proj | Rand Smooth | Rand Smooth Block | Rand Ablation | Adv Train | Block Adv Train | Proposed |
|---|---|---|---|---|---|---|---|---|
| # 7 miss | 84.2± 1.0 | **86.2**± 1.0 | 84.4± 0.9 | 84.4± 0.8 | 82.4± 3.7 | 84.4± 3.8 | 82.6± 2.5 | 86.1± 0.8 |
| #7,8 miss | 83.2± 0.8 | 85.5± 1.5 | 83.4± 0.4 | 83.4± 0.5 | 80.6± 3.0 | 84.4± 4.0 | 84.0± 1.1 | **85.5± 0.3** |
| #1,7 miss | 78.9± 3.1 | 81.5± 0.7 | 79.8± 2.3 | 79.7± 2.4 | **82.4**± 3.7 | 82.3± 2.0 | 75.0± 10.6 | 82.2± 2.0 |

Lastly, we do ablation studies by testing the Clean Accuracy of each defense method, which is the classification accuracy when there is no attack. Table 8 shows the Clean Accuracy of each defense on ExtraSensory dataset. All the methods perform well when there is no attack. The manifold projection method and the proposed method have even higher Clean Accuracy than the Unsecured method. This is likely due to the help of the learned feature subspace.[2]

Table 8: Clean Accuracy (%) of each defense on ExtraSensory dataset, where there is no attack.

| Unsecured | Manifold Proj | Rand Smooth | Rand Smooth Block | Rand Ablation | Adv Train | Block Adv Train | Proposed |
|---|---|---|---|---|---|---|---|
| 83.5 ± 0.7 | **86.2** ± 0.8 | 83.6 ± 0.6 | 83.7 ± 0.5 | 82.2 ± 2.9 | 84.9 ± 3.7 | 80.3 ± 6.4 | 85.7 ± 0.4 |

## C   Implementation details

### C.1   Datasets and computing resources

ExtraSensory dataset (Vaizman et al., 2017) (License CC BY-NC-SA 4.0) contains the measurements from diverse sensors of smart phone and smart watch. We divide the sensors into 10 agents, which correspond to phone accelerometer, gyroscope, magnetometer, watch accelerometer, compass, location, audio, phone states (e.g., battery, wifi, ringing mode), environment sensing (e.g., light, air pressure, humidity, and temperature), and time-of-day (e.g., morning). There are naturally missing-feature problems, e.g., the user may not permit the use of microphone and/or location. The original purpose of this dataset is to recognize the behavioral context of the user (e.g., sitting, walking, and running) and recommend more appropriate music. We use first 1721 samples from a user for training, and the rest 465 samples for testing, with the binary label 'sitting' or not. We split 221 training samples as validation set to tune the hyperparameters of each method. After obtained the hyperparameters, the models are re-trained on all the 1721 training samples.

In NUS-WIDE dataset (Chua et al., 2009), each sample has 634 image features, 1000 text features, and 5 different labels, *i.e.*, 'buildings', 'grass', 'animal', 'water', 'person'. We split the features into 4 agents, where the 1st agent holds 360-d image features, 2nd agent holds the rest of the image features, and the remaining 2 agents each hold 500-d text features. We use 60000 samples for training, 1000

---

[2]It is well-known that subspace projection methods like PCA can help clean the small inlier noise on the sensor measurements and lead to better performance. While robust subspace methods like Robust PCA are not necessary, since there are no large corruptions.

samples for testing targeted attack, and 10000 samples for testing untargeted attack. We split 10000 training samples as validation set to tune the hyperparameters of each method. After obtaining the hyperparameters, the models are re-trained on all the 60000 training samples.

These two datasets are publicly available. Especially the license of ExtraSensory dataset implies it is free to copy, redistribute, remix, transform, and build upon that dataset. The license of the NUS-WIDE dataset is unknown, but we have signed the Agreement and Disclaimer Form of this dataset for use. The datasets do not contain personally identifiable information or offensive content. In particular, the dataset owner of ExtraSensory has removed the personally identifiable information before releasing this dataset.

The experiment was run on a Linux machine with Intel® Xeon® Gold 6132 CPU and 8 NVidia® 1080Ti GPUs. Our code is based on Tensorflow 2.0. The training and testing source codes, as well as the pre-processed ExtraSensory dataset can be found in this GitHub link: `https://github.com/anonymous-github-account-Neurips2022/CoPur.git`. Using one GPU, running the whole training procedure takes a couple of hours on NUS-WIDE dataset and less than an hour on ExtraSensory dataset. Testing each sample takes a few seconds.

## C.2 Implementation details of each method

**Unsecured**: We use the synchronized version of Algorithm 1 in Chen et al. (2020) to jointly train the local feature extractors of each agent as well as FC's global model. More specifically, all the $M$ agents connect to the FC to jointly train their local feature extractors $f_i$ parameterized by $\theta_i, i = 1, ..., M$. In $t$-th iteration, every agent $i$ sends embedded features $\boldsymbol{h}_i^{(j)} \triangleq f_i(\boldsymbol{x}_i^{(j)}; \theta_i^{(t)})$ to the FC. The FC receives $\{\boldsymbol{h}_1^{(j)}, \boldsymbol{h}_2^{(j)}, ..., \boldsymbol{h}_M^{(j)}\}$ from all the $M$ agents, and concatenates them into a long column vector $\boldsymbol{h}^{(j)} \triangleq [\boldsymbol{h}_1^{(j)}; \boldsymbol{h}_2^{(j)}; ...; \boldsymbol{h}_M^{(j)}]$. Then the FC uses $\boldsymbol{h}^{(j)}$ and corresponding label $y^{(j)}$ to update its global model $f_{\theta_0}$ (whose input dimension equals that of $\boldsymbol{h}^{(j)}$) with learning rate 0.005 and batch size 64. After that, for each $i$, the FC calculates $\partial\ell/\partial\boldsymbol{h}_i^{(j)}$ and sends it back to agent $i$. Each agent $i$ then updates its local feature extractor parameterized by $\theta_i$ using the combined gradient $\frac{\partial\ell}{\partial\boldsymbol{h}_i^{(j)}}\frac{\partial\boldsymbol{h}_i^{(j)}}{\partial\theta_i}$ with learning rate 0.005 and batch size 64. The local feature extractors are trained 100 epochs. The dimension of the embedded features of each local feature extractor is set to be 60 on NUS-WIDE dataset, and set to be 32 on ExtraSensory dataset. All other methods use the same local feature extractors as the Unsecured method.

**Oracle**: using the same local feature extractors and global model as the Unsecured method, but the testing instance is uncorrupted (no attack).

**Randomized Smoothing** (Cohen et al., 2019), it uses the same local feature extractors and global model as the Unsecured method. It adds randomly generated Gaussian noise to $\boldsymbol{h}^{test}$ for the FC to make the prediction, the final prediction is by majority voting over 1000 such random trials.

**Randomized Smoothing Block**: the only difference with Randomized Smoothing is that it *knows* which blocks of $\boldsymbol{h}^{test}$ are corrupted, and only adds random Gaussian noise to the corrupted blocks.

**Randomized Ablation** (Levine & Feizi, 2020), the FC learns to make predictions based on randomly selected $K$ agent/block features during the training; and in the inference phase, FC randomly selects $K$ agents to make a prediction. The final prediction is by majority voting over $T$ random trials. However, it's hard to design a base classifier that is able to train on randomly ablated input features. For images, the authors designed a color channel approach to encode the absence of information (*i.e.*, NULL) at ablated pixels: the original 3 channels (red; green; blue) are augmented to (red; green; blue; 1-red; 1-green; 1-blue), while NULL is encoded as (0; 0; 0; 0; 0; 0). The CNN-based classifier can be trained on such data, however, it still does not know such encoding scheme of NULL. It's not clear how to represent the NULL for more general data (e.g., audio) for training a CNN-based classifier. Nevertheless, we use zero value for the ablated features to make it work, for comparison purposes. We tuned $K = 2$ for NUS-WIDE dataset with 4 agents, and $K = 4$ for ExtraSensory dataset with 10 agents.

**Adversarial Training (AT)** (Madry et al., 2018): standard adversarial training via using $T$ iterations of PGD attacks with step-size $\eta$ on $\boldsymbol{h}^{train}$ to generate adversarial examples, but without using projection step to limit the magnitude of adversarial perturbations. For defending against the

targeted attack, it *knows the target label* and leverages that to generate targeted adversarial examples. For defending against untargeted attack, it uses untargeted PGD attack to generate adversarial examples. On NUS-WIDE dataset with 4 agents, we tuned $\eta = 0.1, T = 50$ for targeted attack and $\eta = 0.1, T = 10$ for untargeted attack. On ExtraSensory dataset with 10 agents, we tuned $\eta = 0.5, T = 30$ for targeted attack and $\eta = 0.1, T = 30$ for untargeted attack.

**Block-wise Adversarial Training (BAT)**: since only some block(s) of $\boldsymbol{h}^{test}$ will be corrupted, we further propose and test a block-wise version of AT: for each training sample's embedded features, we randomly select a block and generate adversarial perturbations (via corresponding targeted or untargeted PGD attack) on that block for adversarial training. On NUS-WIDE dataset with 4 agents, we tuned $\eta = 0.1, T = 50$ for targeted attack and $\eta = 0.1, T = 10$ for untargeted attack. On ExtraSensory dataset with 10 agents, we tuned $\eta = 0.5, T = 30$ for targeted attack and $\eta = 0.1, T = 30$ for untargeted attack.

**CoPur**: we use the same local feature extractors as the Unsecured method. After that, as described in Section D, the FC trains an AutoEncoder based on the embedded features of the training samples. The AutoEncoder is a four-layer fully connected neural network with leaky ReLU as the activation function, the dimension of the coding layer is 150. We train 400 epochs with a learning rate of 0.001. Then, FC trains its own global model $f_{\theta_0}$, which has the same architecture as the global model of the Unsecured method, based on the output of the AutoEncoder and the training labels, *i.e.*, $\{\mathcal{D}_\phi(\mathcal{E}_\psi(\boldsymbol{h}^{(j)})), y^{(j)}\}_{j=1}^n$ (200 epochs with learning rate 0.001). We notice that further smoothing the global model via Randomized Smoothing makes little difference in the Robust Accuracy (usually within 1%). The empirical results we reported did not use smoothed model. During the inference, the FC receives the embedded features $\boldsymbol{h}_i^{test}, i = 1, ..., M$ from the agents (not the raw data $\boldsymbol{x}_i^{test}$), and concatenates them into a long column vector $\boldsymbol{h}^{test} \triangleq [\boldsymbol{h}_1^{test}; \boldsymbol{h}_2^{test}; ...; \boldsymbol{h}_M^{test}]$ for feature purification. In feature purification, we aim to solve Eq. 6, i.e., finding the embedded features that approximately lie on the learned feature subspace (or call manifold) while having the smallest objective value. We first try to find a good initial point by simply searching the embedded features on the manifold, i.e., $\min_{l'} \sum_{i \in \Omega} \|[\boldsymbol{h} - \mathcal{D}_\phi(\mathcal{E}_\psi(\boldsymbol{l}'))]_i\|_2$ via gradient descend (the stopping rule is the $\ell_2$-norm of the gradient less than 0.1, or reach 100 iterations). This strategy is motivated by the observation that since the solution $\boldsymbol{l}^*$ of Eq. 6 satisfies its constraint $\sum_{i=1}^M \|[\mathcal{D}_\phi(\mathcal{E}_\psi(\boldsymbol{l}^*)) - \boldsymbol{l}^*]_i\|_2 \le \delta$, by triangle inequality, the objective values of $\sum_{i \in \Omega} \|[\boldsymbol{h} - \boldsymbol{l}^*]_i\|_2$ satisfies $\sum_{i \in \Omega} \|[\boldsymbol{h} - \mathcal{D}_\phi(\mathcal{E}_\psi(\boldsymbol{l}^*))]_i\|_2 - \delta \le \sum_{i \in \Omega} \|[\boldsymbol{h} - \boldsymbol{l}^*]_i\|_2 \le \sum_{i \in \Omega} \|[\boldsymbol{h} - \mathcal{D}_\phi(\mathcal{E}_\psi(\boldsymbol{l}^*))]_i\|_2 + \delta$. Then using $\mathcal{D}_\phi(\mathcal{E}_\psi(\boldsymbol{l}'))$ as the initial point, we apply Lagrange Multiplier method to solve Eq. 6, i.e., $\hat{\boldsymbol{l}} = \arg \min_{\boldsymbol{l}} \sum_{i \in \Omega} \|[\boldsymbol{h} - \boldsymbol{l}]_i\|_2 + \tau \sum_{i=1}^M \|[\mathcal{D}_\phi(\mathcal{E}_\psi(\boldsymbol{l})) - \boldsymbol{l}]_i\|_2$ (the stopping rule is the $\ell_2$-norm of the gradient less than 0.1, or reach $T$ iterations). We notice that 10 iterations are enough, it is not necessary to run too many iterations, as the initial point that we searched on the manifold is quite close to our desired solution (It is also useful to notice that since Eq. 6 allows $\boldsymbol{l}$ approximately lie on the manifold, its objective value will be slightly smaller than $\min_{l'} \sum_{i \in \Omega} \|[\boldsymbol{h} - \mathcal{D}_\phi(\mathcal{E}_\psi(\boldsymbol{l}'))]_i\|_2$). We tuned $\tau$ to be 1000. After that, we feed $\mathcal{D}_\phi(\mathcal{E}_\psi(\hat{\boldsymbol{l}}))$ to FC's trained global model for the prediction.

**Manifold Projection**: It simply projects the corrupted embedded features $\boldsymbol{h}^{test}$ onto the learned manifold via AutoEncoder (the same AutoEncoder as CoPur). Then, it feeds $\mathcal{D}_\phi(\mathcal{E}_\psi(\boldsymbol{h}))$ to FC's trained global model for the prediction (the same global model as CoPur).

# D Training Procedure

While we do not focus on training, we will require an AutoEncoder, trained on uncorrupted data (e.g., during training) that can approximately capture the manifold of the uncorrupted embedded features, *i.e.*, $\sum_{i=1}^M \|[\mathcal{D}_\phi(\mathcal{E}_\psi(\boldsymbol{l}^*)) - \boldsymbol{l}^*]_i\|_2 \le \delta$ for the true embedded features $\boldsymbol{l}^*$ of the testing instance, where $\mathcal{D}_\phi$ is the decoder parameterized by $\phi$, and $\mathcal{E}_\psi$ is the encoder parameterized by $\psi$. Further, we assume that the global model for inference $f_{\theta_0}$ is based on the output of the AutoEncoder and the training labels, *i.e.*, $\{\mathcal{D}_\phi(\mathcal{E}_\psi(\boldsymbol{h}^{(j)})), y^{(j)}\}_{j=1}^n$. Here we provide an example of how to train such a model and the AutoEncoder.

During training, there are a set of $n$ training instances $\{\boldsymbol{x}_1^{(j)}, ..., \boldsymbol{x}_M^{(j)}, y^{(j)}\}_{j=1}^n$. The FC holds the label $y^{(j)}$, while agent $i$ holds partial data $\boldsymbol{x}_i^{\{j\}}$ of the $j$-th instance $\boldsymbol{x}^{(j)} \triangleq [\boldsymbol{x}_1^{(j)}; ...; \boldsymbol{x}_M^{(j)}]$. Each

agent $i$ has its own local feature extractor $f_i$ parameterized by $\theta_i$ that maps the original data vector $\boldsymbol{x}_i^{(j)}$ to embedded feature vector $\boldsymbol{h}_i^{(j)} \triangleq f_i(\boldsymbol{x}_i^{(j)}; \theta_i)$.

**Feature subspace learning**: The FC receives $\{\boldsymbol{h}_1^{(j)}, \boldsymbol{h}_2^{(j)}, ..., \boldsymbol{h}_M^{(j)}\}$ from all the $M$ agents, and concatenates them into a long column vector $\boldsymbol{h}^{(j)} \triangleq [\boldsymbol{h}_1^{(j)}; \boldsymbol{h}_2^{(j)}; ...; \boldsymbol{h}_M^{(j)}]$. Therefore, for each instance $\boldsymbol{x}^{(j)}$, the FC holds corresponding embedded features $\boldsymbol{h}^{(j)}$. Let $\boldsymbol{H} = [\boldsymbol{h}^{(1)}, ..., \boldsymbol{h}^{(n)}]$, where the $j$-th column of $\boldsymbol{H}$ corresponds to $\boldsymbol{h}^{(j)}$.

Based on above $\boldsymbol{H}$, the FC can train an AutoEncoder by minimizing the following objective:

$$\min_{\phi, \psi} \|\boldsymbol{H} - \mathcal{D}_\phi(\mathcal{E}_\psi(\boldsymbol{H}))\|_F \tag{60}$$

where $\mathcal{D}_\phi$ is the decoder parameterized by $\phi$, and $\mathcal{E}_\psi$ is the encoder parameterized by $\psi$.

**Fusion center training**: The fusion center trains its own global model $f_{\theta_0}$ based on the output of the AutoEncoder and the training labels, *i.e.*, $\{\mathcal{D}_\phi(\mathcal{E}_\psi(\boldsymbol{h}^{(j)})), y^{(j)}\}_{j=1}^n$, with learning rate $\eta_0$ and batch size $B_0$.

So far we assumed each agent's local feature extractor is fixed/given. Depending on the applications, the local feature extractors may be trainable. If they are trainable, one can adopt Algorithm 1 of Chen et al. (2020) where FC/server uses its training label to help agents jointly train their local feature extractors, in either synchronized or asynchronous way.

# E  Discussions

## E.1  Why CoPur can do better?

CoPur not only empirically significantly outperforms the baselines, but also has certified robustness guarantees. Interested readers may wonder why CoPur can do better than the baselines. The key lies in its utilization of the block-sparse nature of adversarial perturbations on the feature vector, as well as the underlying redundancy across the embedded features. The closely related Manifold-Projection method utilizes the underlying redundancy across the embedded features, but does not make use of the block-sparse nature of the adversarial perturbations. While the Randomized Smoothing and Adversarial Training type methods do not explore the redundancy across the embedded features, nor the block-sparse nature of adversarial perturbations. Randomized Ablation method implicitly utilizes the block-sparse nature of the adversarial perturbations, but does not explore the underlying redundancy across the embedded features.

## E.2  Limitations of this work

The fraction $(q/M)$ of corrupted blocks that the proposed feature purification method can tolerate not only depends on the index set of observed blocks $\Omega$, but also depends on the underlying feature subspace. This is intuitive, for example, if every agent holds exactly the same embedded features (huge redundancy), then every block of $\boldsymbol{v} = \boldsymbol{r}' - \boldsymbol{r}''$ (defined in Theorem 2) is the same, and $q$ can approach $50\% \times |\Omega|$. However, it's not straightforward to calculate $q$ for the general case. The required condition $\sum_{i \in S_\Omega} \|\boldsymbol{v}_i\|_2 > \sum_{i \in \bar{S}_\Omega} \|\boldsymbol{v}_i\|_2$ in Theorem 2 can be looked as the non-linear extension of the Range Space Property (RSP) in robust linear regression (Flores, 2015; Liu et al., 2018), where the RSP is analogy to another well-known Null Space Property (NSP) in compressive sensing literature. NSP is a fundamental property in compressive sensing, but hard to compute. Fortunately, RSP can be computed by some algorithm (Sharon et al., 2009), though the complexity is very high.

As discussed in Remark 2, we consider a typical case where there is no non-linear activation function in the last layer of the AutoEncoder (other layers can have non-linear activation functions). Let $\boldsymbol{W}_{end}$ be the weight matrix of the last layer of the AutoEncoder. Note that the difference between any two vectors from the range of the decoder $\mathcal{D}_\phi(\cdot)$ is contained in the $Range(\boldsymbol{W}_{end})$, *i.e.*, $\boldsymbol{v} \in Range(\boldsymbol{W}_{end}) \backslash \boldsymbol{0}$. We extended the so called 'leverage constant' for RSP in robust linear regression (Flores, 2015; Liu et al., 2018) to the block-wise 'incomplete' leverage constant $c_{\Omega_q}(\boldsymbol{W}_{end})$ in order to help further characterize our required condition. For example, a sufficient condition for our required condition $\sum_{i \in S_\Omega} \|\boldsymbol{v}_i\|_2 - \sum_{i \in \bar{S}_\Omega} \|\boldsymbol{v}_i\|_2 > 0$ to hold is $c_{\Omega_q}(\boldsymbol{W}_{end}) > 0.5$. Currently we can only calculate some extreme cases, e.g., if every agent holds exactly the same embedded features, and $q = 0.1 \times |\Omega|$, then $c_{\Omega_q}(\boldsymbol{W}_{end}) = 0.9$. One of our future directions is to extend Algorithm 1 of

Sharon et al. (2009) to calculate such block-wise 'incomplete' leverage constant, which is promising, but non-trivial to extend.

### E.3 Motivation for feature-flipping attack

The proposed feature recovery method (CoPur) utilizes the underlying manifold of the uncorrupted features, which is also utilized by several Manifold-Projection based defense methods. This motivates us to design the distributed feature-flipping attack where the corruption vector $e$ aims at introducing correlations with the underlying feature manifold (see detailed discussion in Section 5.1). Fortunately, the proposed defense is still certifiably robust to this attack, owing to making use of the block-sparse structure of the corruption vector $e$. Another motivation for proposing this feature-flipping attack is to make the attack more realistic in practice, which is agnostic to the Fusion Center's global model, label, gradients, as well as the data and local models held by other agents. Since in a collaborative inference setting, it is difficult for the malicious agent to know this information.

### E.4 Broader Impact

This work focuses on defense, which has a positive societal impact. During our study of the weakness of the existing Manifold Projection based defense, we proposed and tested a distributed feature-flipping attack that can successfully attack the baseline defense methods. Our motivation is to use it as a security checking tool for defense methods, though it is possible to be used maliciously. Nevertheless, we further proposed our new defense method CoPur that successfully defends against this attack, both in theory and practice, to eliminate the potential negative societal impact.