# OpenReview forum: "CoPur: Certifiably Robust Collaborative Inference via Feature Purification"
_NeurIPS.cc/2022/Conference — NeurIPS 2022 Accept_

### Official Review · Reviewer_h8UV · 2022-07-09

**Rating:** 5
**Confidence:** 2
**Soundness:** 3 good
**Presentation:** 1 poor
**Contribution:** 3 good

**Summary:**

This paper considers the problem of robust collaborative inference. This paper proposes a feature purification method for this problem. With the block-sparse structure, the authors propose a non-linear decomposition method to identify the corrupted agents. Experiments on the two datasets show the effectiveness of the proposed method.

**Questions:**

Please see weaknesses.

**Ethics Review Area:**

["I don’t know"]

**Limitations:**

The authors discuss the limitations in Appendix. I have no other suggestions.

**Strengths And Weaknesses:**

Firstly, I have to say that I’m not an expert in this area. The review could be misleading.

## Strengths:

1.This paper considers a practical and interesting problem, what if some agents are attacked and send corrupted features to the server.

2.This paper proposes a non-linear decomposition method for identifying poison agents, which is reasonable and should be effective.

## Weakness:

### Originality:

1.I want to know if this paper is the first time to study the problem of the robust collaborative inference, where there are both arbitrary agents and adversarial agents. The arbitrary agents are easy to identify. However, I’m afraid the proposed method achieves a similar performance to identify the adversarial agents compared with baselines.

2. From Eq.(5), the framework aims to find a combined feature $\boldsymbol{l}$ which is on the manifold and is near $\boldsymbol{h}$. The manifold projection could get a similar results for the adversarial sub-features. Could the authors discuss more about it?

### Writtings:
1.After so many times of reading, I guess I understand this paper. The authors introduce their method in Section 2.3, which is very simple. However, it relies block-sparse structure which is detailed stated in Section4. This could cause confuse when understanding the proposed method.

2.The notations are confusing. For example, $\boldsymbol{h}$ and $\boldsymbol{l}$ both denote the feature. Why not use a letter (or with its variants)?

3. the citation format may be ICLR rather NeurIPS.

### Theoretical analysis:
1.This paper provides an extensive theoretical analysis. In fact, I suggest the authors discuss more what the analysis means. Compared with baselines, why CoPur could do better.

2.Could the authors give an intuitive explanation about the effect of the sparsity $\alpha$ on CoPur?


### Experiments:

1.From the ablation studies, CoPur achieves a better performance compared with the manifold projection, what if there are different $\Omega^{c}$ and different $\Omega_{adv}$?

2.More analysis is helpful, for example, The comparison on optimization efficiency.

---

> ### Author Response · Authors · 2022-08-02
> **Thanks for your helpful feedback. Below are our responses:**
>
> --Originality 1: if this paper is the first time to study the problem of the robust collaborative inference, where there are both arbitrary agents and adversarial agents.
>
> R: We assume you refer ‘arbitrary agent’ to the agent that performs the missing-feature attack. To the best of our knowledge, we are the first one to study the robust collaborative inference where there are both agents that performing adversarial attacks and agents performing missing-feature attack.  Our goal is not to identify adversarial agents, but to robustly recover the underlying true features $l$ with theoretical guarantees. None of the baselines has such theoretical guarantee. And none of the baselines can identify adversarial agents.
>
> --Originality 2: From Eq.(5), the framework aims to find a combined feature l which is on the manifold and is near h. The manifold projection could get a similar results for the adversarial sub-features. Could the authors discuss more about it?
>
> R: First to clarify that the true feature $l$ is not necessarily near $h$ (in Euclidean distance), as the corruption $e=h-l$ is allowed to be very large in collaborative inference setting. The objective of Eq.(5) uses L_2,1 norm to encourage the corruption $e$ to be block-sparse, and can guarantee the **exact** recovery of the true feature $l$ (Thm 2). In a sharp contrast, as we discussed in ‘Manifold Projection vs. Feature Purification’ paragraph of Section 4, Manifold Projection usually can **not** recover the true features $l$. *Even worse*, its projected features can be arbitrarily far from the true features $l$, when the magnitude of corruption $e$ gets larger.
>
> --Writing 1:  The authors introduce their method in Section 2.3, which is very simple. However, it relies block-sparse structure which is detailed stated in Section4.
>
> R: Thanks for letting us know. We have moved the block-sparse structure description to Section 2.3 to make it clear.
>
> --Writing 2: h  and L  both denote the feature. Why not use a letter (or with its variants)?
>
> R: Thanks for the suggestion. We can use $\tilde{l}$ instead of $h$ to denote the corrupted version of $l$. (we prefer not to update the notation right now, to avoid confusing other reviewers.)
>
> -- Writing 3: the citation format may be ICLR rather NeurIPS.
>
> R: Do you mean the use of \citet ? We did follow the citation format in Section 4.1 of “Formatting Instructions For NeurIPS 2022” pdf style file.
>
> --Analysis 1: This paper provides an extensive theoretical analysis. In fact, I suggest the authors discuss more what the analysis means. Compared with baselines, why CoPur could do better.
>
> R: Thanks for your suggestions. We are happy to further extend our discussions about what our theoretical analysis means in the remarks (currently it’s somehow constrained by the 9-page limit).
> Compared with baselines, CoPur not only empirically significantly outperforms the baselines, but also has certified robustness guarantees. The key lies on its utilizations of the block-sparse nature of adversarial perturbations on the feature vector, as well as the underlying redundancy across the embedded features. The closely related Manifold-Projection method utilizes the underlying redundancy across the embedded features, but does not make use of the block-sparse nature of the adversarial perturbations. While the Randomized Smoothing and Adversarial Training type methods do not explore the redundancy across the embedded features, nor the block-sparse nature of adversarial perturbations. Randomized Ablation method implicitly utilizes the block-sparse nature of the adversarial perturbations, but does not explore the underlying redundancy across the embedded features.

---

> ### Author Response · Authors · 2022-08-02
> **Responses to other questions**
>
>
> --Analysis 2: Could the authors give an intuitive explanation about the effect of the sparsity α on CoPur?
>
> R: Sure, α is the total fraction of the malicious agents that performs adversarial attack and missing-feature attack. At a high level, CoPur guarantees that it can robustly recover the underlying features $l$ as long as α is small. Let us further explain it using the noiseless case:
>
>  First, Thm 1 studies the case that all the malicious agents perform the missing-feature attack (Thread Model B), that are indexed by $Ω^c$. While Ω is the index set of the observed agents, which are all benign. We have $|Ω^c|=αM$ and $|Ω|=(1-α)M$. Intuitively, the larger the observed set Ω, the required condition in Thm 1 has better chances to be satisfied. While the larger the observed set Ω corresponds to the smaller α.
>
> Second, Thm 2 studies the cases that some malicious agents perform the missing-feature attack (indexed by $Ω^c$, can be an empty set) and some malicious agents perform adversarial attacks (whose number is $\sum_{i\in \Omega} {\unicode{x1D7D9} [ e^*_i \neq  0 ]}$) . So we have
>
> $|Ω^c|+ \sum_{i\in \Omega} {\unicode{x1D7D9} [ e^*_i \neq  0 ]} = αM.$
>
>  As the Fusion Center knows which agents do not provide features, so it knows $|Ω^c|$. Then Thm2 establishes that as long as $\sum_{i\in \Omega} {\unicode{x1D7D9} [ e^*_i \neq  0 ]}$ is small enough (<= q), it can recover the underlying features $l$ exactly.
> The corresponding guarantees for the noisy case can be found in Thm 3 and Corollary 1 of the supplementary. Please let us know if you would like further explanations during the Author- Reviewer Discussion period.
>
> --Experiments 1: From the ablation studies, CoPur achieves a better performance compared with the manifold projection, what if there are different Ωc  and different Ωadv?
>
> R: We just tried several different combinations of Ωadv with Ωc, CoPur consistently performs better than Manifold-Projection. Please refer to Figure 4 in the updated supplementary.
>
> --Experiments 2: More analysis is helpful, for example, The comparison on optimization efficiency.
>
> R: Thanks for the suggestion, we are happy to add more analysis perhaps to the supplementary. Let us briefly mention the efficiency of some defense strategies. Both CoPur and Manifold-Projection need to train/use an AutoEncoder. CoPur additionally needs to use Lagrange Multiplier method to solve its objective Eq.(6) via gradient descent during the inference. The Randomized Ablation method needs to train a classifier that learns to make predictions based on randomly selected K agent/block features. During its inference, it randomly selects K agent/block features to make a prediction, and repeats this for T random trials. There are a combinatorial number of such selections, which will blow up quickly when the total number M of the agents and K get larger. Adversarial Training and its variants need to run multiple iterations of gradient descent on each training sample to generate adversarial example for training.
>
> We hope this addresses your concerns and can help you re-evaluate our work. Please let us know if you still have concerns during the Author- Reviewer Discussion period.

---

> ### Author Response · Authors · 2022-08-09
> **Any remaining concern？**
>
> Dear Reviewer h8UV, we hope our responses have addressed your concerns. Please kindly let us know if there is anything not clear or if you have remaining concerns.

---

> > ### Comment · Reviewer_h8UV · 2022-08-09
> > **Reply to the authors**
> >
> > Thanks for the detailed response. Most of my concerns are about the presentation and they are addressed from the response. I would like to keep the **Borderline accept** score.

---

> > > ### Author Response · Authors · 2022-08-09
> > > **Thanks for your feedback on our response!**
> > >
> > > Thanks for your feedback on our response! We would appreciate if you can further consider the merit of a certifiable defense to solve a trendy distributed/collaborative inference problem, as well as the impact of theoretical innovation on separating low-dim manifold and block-sparse structure. We believe our work is timely, and has high potential impact. Hope you will consider raising your score above a Borderline Accept during the potential deeper 'Reviewers Discussion' stage.

---

### Official Review · Reviewer_vVL3 · 2022-07-10

**Rating:** 4
**Confidence:** 4
**Soundness:** 2 fair
**Presentation:** 3 good
**Contribution:** 2 fair

**Summary:**

This paper proposed a certifiably robust COllaborative inference framework via feature PURification (CoPur), by leveraging the block-sparse nature of adversarial perturbations on the feature vector, as well as redundancy across the embedded features (by assuming the overall features lie on an underlying lower dimensional manifold). This paper also gave theoretical results showing that the proposed method can robustly recover the true feature vector.  Extensive experiments showed the effectiveness of the proposed method. Besides, an untargeted distributed feature-flipping attack was proposed.


**Questions:**

Q1: The main contribution (Eq.6) works out based on the prior knowledge that adversarial feature $e$ has a block-sparse structure. However, theoretical results didn't formalize the sparsity of $e$ clearly enough.

Q2: The proposed feature-flipping attack is not well-evaluated. Experiments showed that the proposed defense has better accuracy against this attack compared to other defenses. However, no experimental results are given to indicate that this attack is stronger than other attacks.

Q3: How sparsity of adversarial feature $e$ affects the robust accuracy is not evaluated.

Q4: Figure 2 shows that when amplification is low, CoPur has higher robust accuracy than Oracle. Can you explain why this phenomenon happens? If CoPur is applied to unattacked features, will it improve the standard accuracy?

Q5: Why the solution to Eq.6 is not mentioned in the main text?


**Limitations:**

No limitations discussed in the paper. See questions.


**Strengths And Weaknesses:**

Pros
1. Theoretical analysis is given, showing that the recovered benign feature is close to the true feature with bounded error.
2. Feature-flipping attack is proposed to demonstrate the weakness of manifold projection defense.

Cons
1. Theoretical results are not adequate.
2. Lack of experimental results.

---

> ### Author Response · Authors · 2022-08-02
> **Thanks for your constructive feedbacks! Below are our responses**
>
> --Cons 1 & Q1: The main contribution (Eq.6) works out based on the prior knowledge that adversarial feature  has a block-sparse structure. However, theoretical results didn't formalize the sparsity of  clearly enough.
>
> R: In the noiseless case, the sparsity of e is clearly formalized in Thm 2 as $\sum_{i\in \Omega} {\unicode{x1D7D9}[ e^*_i \neq  0}]\leq q$.
>
> For the noisy case, we can easily have a corollary of Thm 4 reads as follows (the proof is essentially the same as Thm4):
>
> **Corollary 1** (Stable feature recovery under threat model C): Assume the trained AutoEncoder satisfies $\sum_{i=1}^M ||[\mathcal{D_{\phi}(\mathcal{E_{\psi}}} ({l}^*))-{l}^*]_i ||_2 \leq \delta$ for the underlying uncorrupted feature vector ${l}^*$. Let $ \hat{ l}$ be the solution of Eq. 6.  Given the incomplete observations
>
> $h_{\Omega}=l_{\Omega}^*+e^*_{\Omega},$
>
> where $\Omega$ is the index set of observed agent blocks. For $\forall  r',  r''$ in the range of the decoder $\mathcal{D_{\phi}}(\cdot)$ where $r' \neq r''$, define $ v= r'- r''$, if for any partition {$\{S_{\Omega},\bar{S}_{\Omega}\}$}
>
> of $\Omega$ with $| \bar{S}_{\Omega} |=q$,
>
> it holds that $\sum_{ i \in S_{\Omega}} || v_i ||_2 $
>
> $ -  \sum_{ i \in \overline{S}_{\Omega}} || v_i ||_2 > 2\delta$ for $\forall  || v ||_2 > \Delta$,
>
> then $||\mathcal{D_{\phi}(\mathcal{E_{\psi}}} (\hat{ l}))-\mathcal{D_{\phi}(\mathcal{E_{\psi}}} ( l^*)) ||_2 \leq \Delta$
>
> as long as $\sum_{i\in \Omega} {\unicode{x1D7D9} [ e^*_i \neq  0 ]}\leq q$.
>
>
> which would clearly formalize the sparsity of $e$. We temporarily add this corollary to Section A of the supplementary (in blue color) due to space limit. Thanks for your suggestion!
>
> --Q2: The proposed feature-flipping attack is not well-evaluated. Experiments showed that the proposed defense has better accuracy against this attack compared to other defenses. However, no experimental results are given to indicate that this attack is stronger than other attacks.
>
> R: We don’t expect this attack to be stronger than some other attacks like PGD attack, since it requires no information at all, just flipping the features. Actually Table 1 and Figure 2 already show that PGD attack is stronger, as it requires much more information, e.g., server’s global model, label, gradients, etc. We introduce this attack for two reasons: First, to adaptively attack Manifold-Projection method and our method. This attack aims at introducing correlations with the underlying feature manifold (see discussion in page 8, line 358), which both Manifold-Projection method and our method rely on; Second, it is well-aligned with our use case and may be considered more realistic for collaborative inference  – because it is agnostic to the model, training data, label, gradients, as well as the testing data and local models held by other agents. This matches practice since in real applications, the malicious agent may not know the Fusion Center’s global model and gradients, as well as the data and local models held by other agents.
>
> --Cons 2 & Q3: How sparsity of adversarial feature e  affects the robust accuracy is not evaluated.
>
> R: Thanks for raising this question, but we did evaluate that, e.g., see Figure 4 of the supplementary (which is Figure 5 in the updated supplementary). Sorry that we put it in the Appendix due to space limit, and we are happy to move it to the main text, to further demonstrate that the proposed method is significantly more robust than baselines.
>
> --Q4: Figure 2 shows that when amplification is low, CoPur has higher robust accuracy than Oracle. Can you explain why this phenomenon happens? If CoPur is applied to unattacked features, will it improve the standard accuracy?
>
> R: Yes, that’s the beauty of the subspace/manifold based methods! In Table 8 of the supplementary, both Manifold Projection method and CoPur improve the standard accuracy of oracle on the unattacked features. Because even if there is no attack, the data of various IoT sensors (like in ExtraSensory) are typically noisy (small inlier noise). It is well understood that the subspace denoising method can help improve IoT data, e.g., Sanyal & Zhang (2018).  If you are familiar with face recognition, the famous PCA subspace based method (Eigenface) also improves face recognition accuracy.
>
> Sanyal & Zhang (2018): S. Sanyal and P. Zhang, "Improving Quality of Data: IoT Data Aggregation Using Device to Device Communications," in IEEE Access, vol. 6, pp. 67830-67840, 2018.
>
> --Q5: Why the solution to Eq.6 is not mentioned in the main text?
>
> R: We moved it to the Appendix due to the space limit. We are happy to move it back to the main text.
>
>
> We hope this addresses your concerns and can help you re-evaluate our work. Please let us know if you still have concerns during the Author- Reviewer Discussion period.

---

> ### Author Response · Authors · 2022-08-09
> **Any remaining concern？**
>
> Dear Reviewer vVL3, we hope our responses have addressed your concerns. Please kindly let us know if there is anything not clear or if you have remaining concerns.

---

### Official Review · Reviewer_mjnb · 2022-07-10

**Rating:** 6
**Confidence:** 4
**Soundness:** 3 good
**Presentation:** 3 good
**Contribution:** 3 good

**Summary:**

This work focuses on feature fusion in distributed machine learning scenarios. In this work, a distributed feature reconstruction method is proposed to recover noise / incomplete features committed by different local clients. The main idea is to decompose observed uncorrupted features into the underlying noise-free features and additional block-wise sparse noise corruption components. After that, a L2,1 signal recovery method is applied to estimate the noise-free feature profiles by enforcing the L2,1 norm sparsity regularisation over the estimated noise components. In parallel, the feature recovery process is accompanied with the learning of an AutoEncoder architecture performed over the recovered feature vector. The goal is to generate a smoothed feature representation from the recovered features for the final classification step. Theoretical and experimental study demonstrate the provably effectiveness of the proposed feature reconstruction method.

**Questions:**

Please find the questions in the second and third points in the listed weakness.

**Ethics Review Area:**

["I don’t know"]

**Limitations:**

Please find the listed weakness points.

**Strengths And Weaknesses:**

Strong points:

1.  Recovering the noise-free feature profiles provides less noisy information for classification, compared to aligning noisy features to the subspace of normal data.
2.  It is decent to make use of the block-wise sparsity structure of the noise components and formulate the feature recovery problem as the L2,1-regularized regression process.
3. Theoretical study established for the noisy case helps clarify the practical implication of the proposed method.

Weak points:
1. Though it is interesting to see a novel feature-corruption-based attack method is proposed, the link between designing this attack algorithm and the feature recovery method (CoPur) is not clearly explained.
2. How does the proposed CoPur method perform agains the distributed feature-flipping attack? Is it the combined attack in Table.2?
3. How many malicious agents can the proposed CoPur defense method tolerate ? It is related to the break point of the proposed method.

---

> ### Author Response · Authors · 2022-08-02
> **Thanks for your helpful feedbacks! Below are our response to your concerns:**
>
> --Weak point 1: Though it is interesting to see a novel feature-corruption-based attack method is proposed, the link between designing this attack algorithm and the feature recovery method (CoPur) is not clearly explained.
>
> R: The proposed feature recovery method (CoPur) utilizes the underlying manifold of the uncorrupted features, which is also utilized by several manifold-projection based defense methods. This motivates us to design a novel feature-corruption-based attack where the corruption vector $e$ aims at introducing correlations with the underlying feature manifold (see discussion in page 8, line 358). (Fortunately, our proposed defense is still certifiably robust to this attack, owing to making use of the block-wise sparsity structure of the corruption vector $e$, as you correctly pointed out in Strong Points.) Another motivation for proposing this feature-corruption-based attack is to make the attack more realistic in practice, that is  agnostic to the global model, label, gradients, as well as the data and local models held by other agents. Since in collaborative inference setting, it is difficult for the malicious agent to know this information.
>
> --Weak point 2: How does the proposed CoPur method perform against the distributed feature-flipping attack? Is it the combined attack in Table.2?
>
> R: These results are shown in Figure 2 of the main paper (as well as Figure 3, Tables 4 & 6 of the supplementary). You can see that the proposed CoPur method significantly outperforms baseline defenses with a large margin. And more importantly, the proposed CoPur method has certifiable robustness guarantees against this attack.
>
> --Weak point 3: How many malicious agents can the proposed CoPur defense method tolerate ?
>
> R: It depends on the underlying redundancy across the features held by the agents. As we discussed at the end of Remark 1, it can tolerate up to 50%×|Ω| corrupted agents if there is enough redundancy among the agents, e.g., if every agent holds exactly the same embedded features. But one should not expect such a high breakpoint when there is much less mutual information across the features held by the agents. In the Robust Linear Regression literature $y=Ax+e$ [Flores, 2015; Sharon et al., 2009]., the breakpoint of the $\ell_1$
> estimator depends on the property of the underlying linear subspace (range space of matrix $A$), i.e., Range Space Property [Flores, 2015; Liu et al., 2018]. There is no universal value for the breakpoint. While we extend the low-dimensional linear subspace to the more general low-dimensional manifold (that embedded features $l$ lies on), the breakpoint of CoPur naturally depends on the range property of the manifold/Decoder (e.g., see Thm 2 and Corollary 1 of the supplementary). And as we discussed in Remark 1, it can be considered that our analysis provides a non-linear and missing-block extension of the Range Space Property.
>
>
> We hope this addresses your concerns and can help you re-evaluate our work.

---

> ### Author Response · Authors · 2022-08-09
> **Any remaining concern？**
>
> Dear Reviewer mjnb, we hope our responses have addressed your concerns. Please kindly let us know if there is anything not clear or if you have remaining concerns.

---

### Official Review · Reviewer_d75c · 2022-07-15

**Rating:** 5
**Confidence:** 4
**Soundness:** 3 good
**Presentation:** 2 fair
**Contribution:** 3 good

**Summary:**

This paper aims to improve the robustness of the deep learning model for collaborative inference. The authors propose a pre-processing-based defense method CoPur against inference phase attacks. The proposed method first leans the purified embedded features from the observed embedded features by using a generative model.Then, the purified embedded features are fed to the Fusion Center for prediction. This paper provide sufficient theoretical analyses to illustrate that the proposed method is able to recover the underlying true features exactly. Empirical results for targeted and non-targeted attacks are presented in this paper, which is used to demonstrate the effectiveness of the proposed method.

**Questions:**

1. Why are the well-known robust aggregation methods used for HFL not applicable? Please provide some further explanations.
2. The authors say they propose to improve robustness based on the underlying redundancy among the features held by the IoT devices. However，it is not very clear to me how the proposed method obtains and exploits redundant information, is it through an autoencoder?
3. The method proposed by the authors seems to be mainly aimed at off-manifold attacks. Is the proposed method effective against on-manifold attacks [1]?
4. PGD is also suitable for crafting non-targeted adversarial noise, why do the authors not consider the non-targeted PGD? In addition, the Autoattack [2] is a more powerful attack, the authors can use it to futher evaluate the effectiveness of the proposed defense.
5. The poposed method is a pre-processing-based method, but the authors do not compare it with other pre-processing-based adversarial defense methods, such as [3].
6. The authors do not consider the stronger adaptive attack [4]. For example, the used autoencoder may also be attacked. The authors should design adaptive attacks to more comprehensively evaluate the robustness of the proposed method.

[1]. Disentangling adversarial robustness and generalization. CVPR, 2019.
[2]. Reliable evaluation of adversarial robustness with an ensemble of diverse parameter-free attacks. ICML, 2020.
[3]. A self-supervised approach for adversarial robustness. CVPR, 2020.
[4]. Obfuscated gradients give a false sense of security: Circumventing defenses to adversarial examples. ICML, 2018.


-------------------------------------------------------
After rebuttal

The authors solve most of my concerns, so I am willing to modify the score to borderline accept.

**Limitations:**

The authors do not consider some closely related works and some import attacks to further show the effectiveness of the proposed method.

**Strengths And Weaknesses:**

Strengths:
1. This paper proposes a certifiable defense to defense against inference phase attacks for the collaborative inference task. This method is based on the pre-processing strategy, which does not require much computing cost and does not change the original information of agents.
2. This paper provide sufficient theoretical analyses to demonstrate that the proposed method is certifiable. The authors analysis two cases (i.e., noiseless case and noisy case) and use them to illustrate that solving some relaxed objectives is able to recover the underlying true features exactly.
3. The authors design a non-target attack (i.e., distributed feature-flipping attack) against the collaborative inference model to evaluate the effect of the baseline defense models and the proposed defense model. This helps to evaluate the effectiveness of adversarial defenses.

Weaknesses:
1. The authors emphasize that in their considered inference setting, the features obtained by the agents are quiet different and the redundancy is less apparent. However, the authors seems not to introduce how to design a specific mechanism for this problem, and how the proposed method handles the issue of less redundancy in detail. In addition, the authors do not discuss the performances of the proposed method under different redundancy situations.
2. IoT applications are often communication-limited. The authors do not evaluate the communication requirements of the proposed method, and they do not show whether the improvement mechanism would increase the communication requirements.
3. Some closely related works are not discussed, and some closely related methods are not compared. Some import attacks are not used to evaluate the defense.
4. Writing and presentation need improvement. For example, the caption in Fig. 1 does not clearly describe the necessary information in the figure, such as what does the variable represent? In addition, \Omega_{adv} is the index for all corrupted features (\Omega = \Omega_{benign} U \Omega_{adv}), but in Eq. 3, it becomes the index for perturbed features, |\Omega_{adv}| in Eq. 3 should be \alpha.

---

> ### Author Response · Authors · 2022-08-02
> **Thanks for your constructive suggestions! Below are our responses to the weakness/limitations and Questions 1-2**
>
> --Weakness 1. How to use redundancy?
>
> R: First to clarify that we did not mean ‘less redundancy’. We meant there is redundancy which is ‘less apparent’ to utilize. The mechanism is exactly through using a lower-dimensional AutoEncoder that captures such redundancy. Contrast this with HFL where all the features are shared, and samples are often assumed from some well-behaved distribution, so the redundancy is more obvious. Please also see Response to your Question 2. We have discussed the performance of the proposed method under the high redundancy setting at the end of Remark 1. For the general redundancy situations, the tradeoff between redundancy and robustness is characterized through the interplay between the underlying AutoEncoder and the number of malicious agents (e.g., see Thms 2 & 4 and Corollary 1 in the supplementary).
>
>
> --Weakness 2. Communication cost:
>
> R: Thanks for raising this useful point. In our proposed method, each agent only needs to send its embedded features to the Fusion Center for robust collaborative inference, the communication is exactly the same as the non-robust/vanilla method. Thus, our improvement mechanism does not increase communication compared to standard collaborative inference.
>
>
> --Weakness 3 / Limitations: Related works/methods/attack :
>
> R: Thanks for pointing out some related works, we will incorporate them in Section 4. But most of the related methods/attack therein (that you suggested in the Questions) are not applicable in our collaborative inference setting. Please refer to our detailed responses to your questions.
>
>
> --Weakness 4. Writing:
>
> R: Thanks for the suggestion, we will add descriptions to the caption of Fig. 1.  Actually $\Omega_{adv}$ is the index for perturbed features, while $\Omega^c$ is the index for missing features, so the constraint in Eq. 3 is correct. We will make this notation more clear.
>
>
> --Question 1. Why robust aggregation used for HFL not applicable?
>
> R: In HFL every agent proposes an update of the parameters of a common ML model. The server of HFL receives these updates and simply performs a robust averaging. But in our collaborative inference setting, each agent may contribute quite different features,  and the server/FC concatenates them into a long feature vector for classification. The features of each agent may have quite different dimensions. Even if we force them to be the same dimension, simply averaging these  features  (e.g., image features and acoustic features) may not be an effective approach (also note that the averaged features have less representation power than the concatenated features).
>
>
> --Question 2. Exploits redundant information through an autoencoder?
>
> R: Yes, through an autoencoder that captures the underlying lower-dimensional manifold. One intuitive example is multi-view sensing, where each agent holds an image from a slightly different view of an object, say vehicle. There are significant redundancies among the agents and the underlying dimension of the overall multi-view images is much lower than the total number of pixels.
>
> At the end of the Remark 1, we give a toy example where every agent holds exactly the same embedded features (high redundancy), the proposed method CoPur can tolerate nearly 50% malicious agents.
>
> It may be easier to understand when the features of each agent are linear dependent, then the overall concatenated feature vector $l$ would come from a lower dimensional linear subspace. Let us use $l^{(j)}$ to denote the whole concatenated feature vector for j-th sample. The matrix $L=[l^{(1)},...,l^{(n)}]$ for n samples (each column corresponds to a sample) would be low-rank, since there are linear dependencies among its rows.

---

> ### Author Response · Authors · 2022-08-02
> **Below are our responses to your Questions 3-4**
>
> --Question 3. The method proposed by the authors seems to be mainly aimed at off-manifold attacks. Is the proposed method effective against on-manifold attacks [1]?
>
> R: The proposed method is certifiably robust against any attacks with any adversarial perturbations as long as the fraction of the malicious agents is small. The on-manifold attack suggested [1] violates the core assumption. This is because the attack adds perturbations to the latent layer of the AutoEncoder to generate the adversarial example $h$. However, the resulting  adversarial perturbations $e$ on the embedded features $l$ (where $h=l+e$) would be non-zero in most (likley all) agent blocks. As noted this violates the collaborative inference threat model where only a small fraction of the agents are malicious. Further, in our collaborative inference setting, the magnitude of the adversarial perturbations is allowed to be very large (which is different from the centralized setting where the adversarial perturbations are often constrained to be small magnitudes). If the adversary can compromise all the agents, it would simply replace the original embedded features $l$  by  $l^{target}$ of the target class, and no method can defend against such an adversary. The budget constraint of the adversary in our collaborative inference setting is the number of the compromised agents, which limits the adversarial manipulation on the manifold. And the theoretical characterization of the interplay between the number of malicious agents and the underlying manifold can be found in Thm 2, where we show CoPur can recover the underlying embedded features $l$ exactly as long as the number of the malicious agents is small. (The corresponding robust recovery guarantee in the noisy case can be found in Corollary 1 of the supplementary.)
>
> --Question 4. PGD is also suitable for crafting non-targeted adversarial noise, why do the authors not consider the non-targeted PGD? In addition, the Autoattack [2] is a more powerful attack, the authors can use it to futher evaluate the effectiveness of the proposed defense.
>
> R: Non-targeted PGD is less interesting given that our ExtraSensory dataset is binary classification and we have already tested the targeted PGD on that. As the task is binary, both Targeted PGD and non-targeted PGD attacks will seek to flip the label.
>
> On the other hand, our proposed non-targeted distributed feature flipping attack is much more interesting. Since in practice, the malicious agent does not know the Fusion Center’s global model, labels, and gradients, nor the data and local models held by other agents. The proposed distributed feature flipping attack is agnostic to all that information, making it much more feasible for practical use. Furthermore, the proposed distributed feature flipping attack is also shown to be effective in attacking various state-of-the-art defenses.
>
> Regarding the Autoattack [2] that you suggest, it consists of 3 types of attacks:
>
> (1) PGD with automatic chosen step size across iterations: In our experiments, the step size of the PGD attacks are carefully tuned for each dataset (to make the attack strong). It’s arguable whether the automatic chosen step size of PGD [2] would be stronger than our manually tuned one. Also, note that the PGD attack in our experiments is already very successful in attacking many defenses.
>
> (2) Fast Adaptive Boundary Attack (FAB): Its goal is to minimize the norm of the perturbation necessary to achieve a misclassification, which is unnecessary since, in our collaborative inference setting, the perturbation is unbounded.
>
> (3) Square attack: this is a black-box attack that does not have access to the gradient information and is weaker than the PGD attack we tested. It aims at randomly searching the adversarial perturbation at the boundary of the $\ell_p$ norm (e.g., $\ell_\infty$ norm) constraint. Again, this attack is designed under the centralized setting where the magnitude of adversarial perturbation is limited to be small. While in our collaborative inference setting, the adversarial perturbation is unbounded, one can not search the adversarial perturbation at the boundary.
>
> Lastly, if CoPur is an empirical defense, we agree that it should be exhaustively tested against various attacks. And even so, experimental demonstrations of a defense’s efficacy on all currently existing attacks do not provide a general proof of security. However, CoPur has **provable security guarantees**. It’s not necessary to exhaustively test it. Even your suggested adaptive attack paper [4], which aims at evaluating its adaptive attack efficacy against all ICLR 2018 defense methods, “**omit two defenses with provable security claims**” (see Section 5 of [4]).

---

> ### Author Response · Authors · 2022-08-02
> **Below are our responses to your Questions 5-6**
>
> --Question 5. The poposed method is a pre-processing-based method, but the authors do not compare it with other pre-processing-based adversarial defense methods, such as [3].
>
> R: We have compared the proposed method with a pre-processing-based adversarial defense method — Manifold Projection, which is the most closely related pre-processing-based method to ours. Thanks for suggesting other pre-processing method [3], but that method needs to generate the adversarial training examples based on the assumption that the magnitude of the adversarial perturbation $e$ is bounded by small $\epsilon$ (see Eq. 4 of [3]), which is different from our collaborative inference setting where the adversarial perturbation is unbounded. Further, that method aims at learning a neural network P that maps any perturbed feature $h$ to its unperturbed version $l$. However, in our collaborative inference setting, arbitrary α fraction of the blocks of feature $l$ can be perturbed by arbitrarily large values. Based on the high (combinatorial) cardinality of the input space, it is highly doubtful that one can learn a magic neural network P that is able to map a combinatorial number of different block-perturbation patterns (and with arbitrary unbounded perturbation values) to the underlying true feature $l$. In a sharp contrast, our proposed method can guarantee recovering the underlying true feature $l$ exactly.
>
> --Question 6. The authors do not consider the stronger adaptive attack [4]. For example, the used autoencoder may also be attacked. The authors should design adaptive attacks to more comprehensively evaluate the robustness of the proposed method.
>
> R: First to clarify that we do not directly use AutoEncoder in our networks. It is used in our robust-decomposition optimization problem. The proposed feature recovery method (CoPur) relies on underlying feature manifold, so does manifold-projection based defense. Therefore we proposed and tested an **adaptive** attack (distributed feature-flipping attack) where the corruption vector $e$ aims at introducing correlations with the underlying feature manifold (see discussion in page 8, line 358).
>
> Some adaptive attack strategies in [4] implicitly use the assumption that the magnitude of adversarial perturbation is small, which does not hold in our collaborative inference setting. For example, the attack strategy in Section 4.1.1 of [4] approximates defender’s denoising function g(x) by x, which does not hold for CoPur’s denoising mechanism.
>
> Nevertheless, we tested the most suitable reparameterization-based adaptive attack of [4] via PGD against CoPur (which tries to attack the AutoEncoder). It turns out that the Robust Accuracy of CoPur under this adaptive attack only degrades 2% compared to its 79% Robust Accuracy in Table 1 on the ExtraSensory dataset, and only degrades 1.2% compared to its 79.1% Robust Accuracy in Table 2. The Robust Accuracies of CoPur are still significantly much higher than baselines even under this adaptive attack on the ExtraSensory dataset. On the NUS-WIDE dataset, the Robust Accuracy of CoPur only degrades 1.4% compared to its 83.7% Robust Accuracy in Table 1, which still performs better than baselines. We will add the adaptive attack results to Table 1&2 to further convince the readers. Thanks for your suggestions.
>
>
> We hope our responses have addressed your concerns and can help you re-evaluate our work. Please let us know if you still have any concerns during the Author- Reviewer Discussion period.

---

> > ### Comment · Reviewer_d75c · 2022-08-08
> > **Modify the score to borderline accept**
> >
> > Thanks for your response, the answers solve most of my concerns, so I am willing to modify the score to borderline accept.

---

> > > ### Author Response · Authors · 2022-08-08
> > > **Thanks for your feedback on our response!**
> > >
> > > Thanks for your feedback on our response! We are glad that we have solved most of your concerns. Please do not hesitate to let us know if there is anything not clear or if you have remaining concerns. Also thanks for raising the score, please remember to update the score in the system by editing your previous review. We would appreciate if you can further consider the merit of a certifiable defense to solve a trendy distributed/collaborative inference problem, as well as the impact of theoretical innovation on separating low-dim manifold and block-sparse structure.

---

### Official Review · Reviewer_Zg9q · 2022-07-18

**Rating:** 6
**Confidence:** 2
**Soundness:** 3 good
**Presentation:** 4 excellent
**Contribution:** 3 good

**Summary:**

This paper studies robust inference under the vertically split features setting, which appears in applications like e Internet-of-Things (IoT). The paper considers three types of threat models, including distributed adversarial attack, missing feature attack, and combined attack. The paper proposes a sparse recovery-based robust inference algorithm and provides recovery guarantees. The paper provides experimental results to show the superior performance of the proposed method.

**Questions:**

The recovery theorems rely on critical hyperparameters like delta, which is empirically tuned on a validation set according to the supplement. Is it possible to provide some theoretical guidance to choose such hyperparameters so that the certified recovery is always ensured?

**Limitations:**

The authors have adequately addressed the limitations and potential negative societal impact of their work

**Strengths And Weaknesses:**

Strengths:

(1) The problem of robust inference under the vertically split features setting is new and interesting.

(2) The proposed method has recovery theory and better empirical performance.

Weakness:

(1) The idea of sparse recovery-based algorithm design is not new.

---

> ### Author Response · Authors · 2022-08-02
> **Thanks for your constructive feedback, below is our response to your concerns:**
>
> Weakness: (1) The idea of sparse recovery-based algorithm design is not new.
>
> R: We agree that sparse recovery is not new, which has been studied over decades. But sparsity is the nature of this collaborative inference problem that should be utilized. One of our key innovations is the proposing of a non-linear robust decomposition method that decomposes a potentially corrupted and incomplete feature vector into two parts with recovery guarantees: one lies (exactly or approximately) on the underlying feature manifold; and the other with a block-sparse structure. The theoretical guarantees of the separation between non-linear manifold and block-sparse structure are fundamentally new.
>
> Q: Theoretical guidance to choose such hyperparameter:
>
> R: This is a great question. Recall that the AutoEncoder is trained on uncorrupted data (e.g., during training). One can use the autoencoder error, i.e., $\sum_{i=1}^M ||[\mathcal{D_{\phi}(\mathcal{E_{\psi}}} (l))- l]_i ||_2 $ of such uncorrupted data to get a sense of the lower bound and approximate range of  δ. In the testing, δ appears in the constraint
>
> $\sum_{i=1}^M ||[\mathcal{D_{\phi}(\mathcal{E_{\psi}}} (l))- l]_i ||_2 \leq \delta .$
>
> Of course one can set δ very large to meet this constraint and obtain theoretical guarantees of robust recovery of true feature $l$ according to Theorems 3&4. But we want to remind the reader that setting δ too large would result in a large recovery error bound $\Delta$.  For example, Theorem 3 shows that
>
> $ \mbox{if}  \sum_{i \in {\Omega}} || v_i ||_2> 2 \delta \mbox{ for }  \forall   || v ||_2 > \Delta,$
>
> $$ \mbox{then } || \mathcal{D_{\phi}(\mathcal{E_{\psi}}} (\hat{l}))-\mathcal{D_{\phi}(\mathcal{E_{\psi}}} ( l^*)) ||_2 \leq \Delta. $$
>
> As long as the constraint $\sum_{i=1}^M ||[\mathcal{D_{\phi}(\mathcal{E_{\psi}}} (l))- l]_i ||_2 \leq \delta $  is met, smaller δ leads to smaller recovery error bound  $\Delta$. So we prefer to use a validation set to choose an appropriate value. We will add such discussion in our revision.
>
> We hope this addresses your concerns and can help you re-evaluate our work.

---

> ### Author Response · Authors · 2022-08-09
> **Any remaining concern？**
>
> Dear Reviewer Zg9q, we hope our responses have addressed your concerns. Please kindly let us know if there is anything not clear or if you have remaining concerns.

---

### Meta-Review · Area_Chair_8JsG · 2022-08-24

**Recommendation:** Accept
**Confidence:** Less certain

**Metareview:**

This paper focuses on improving the robustness of the model for collaborative inference. A pre-processing-based defense method is proposed against inference phase attacks. Both theoretical analyses and empirical evaluations are provided to demonstrate the effectiveness of the proposed method.

In the review process, mixed reviews are achieved. Four reviewers are positive about this paper, while one reviewer is negative. Reviews appreciate the strengths of the paper: (1) the idea is interesting; (2) the pre-processing strategy does not require much computing cost; (3) the theoretical analysis is detailed and solid; (4) empirical evaluations are overall convincing.

One reviewer who is negative about this paper points out major concerns: (1) theoretical results on the sparsity; (2) empirical evaluations on the sparsity; (3) the problems in the presentations of the paper. Major concerns have been addressed during the rebuttal. We suggest the author carefully merge the rebuttals in the final version.

**Award:**

No

---

### Decision · Program_Chairs · 2022-09-14

Accept